# Goal Driven Discovery of Distributional Differences via Language Descriptions

**Ruiqi Zhong,**[*] **Peter Zhang, Steve Li, Jinwoo Ahn, Dan Klein, Jacob Steinhardt**

## Abstract

Exploring large corpora can generate useful discoveries but is time-consuming for humans. We formulate a new task, D5, that automatically discovers differences between two large corpora in a goal-driven way. The task input is a problem comprising a user-specified exploration goal ("*comparing the side effects of drug A and drug B*") and a corpus pair (collections of patients' self-reported reactions after taking each drug). The output is a goal-relevant description (discovery) of how these corpora differ (patients taking drug A "*mention feelings of paranoia*" more often). We build a D5 system, and to quantitatively evaluate its performance, we 1) build a diagnostic benchmark, SYND5, to test whether it can recover known differences between two synthetic corpora, and 2) contribute a meta-dataset, OPEND5, aggregating 675 open-ended problems ranging across business, social sciences, humanities, machine learning, and health. With both synthetic and real datasets, we confirm that language models can leverage user-specified goals to propose more relevant candidate discoveries, and they sometimes produce discoveries previously unknown to the authors, including demographic differences in discussion topics, political stances in speech, insights in commercial reviews, and error patterns in NLP models. Finally, we discuss the limitations of our D5 system, which discovers correlation rather than causation and potentially reinforces biases when misused; therefore, practitioners should treat the outputs of our system with caution.

## 1 Introduction

Exploring large corpora and generating discoveries from them can be ad hoc and laborious. For example, to compare the side effects of drug A and drug B, doctors might inspect two large corpora of patients' self-reported reactions after taking each drug; based on ad hoc insights, they hypothesize that patients taking drug A more often "*mentions feelings of paranoia*", and then validate this hypothesis by laboriously inspecting the two corpora. Since machines can automatically process a large amount of texts, we might hope for ML systems to facilitate exploratory analyses like the one above.

However, an ML task requires a unified input-output space and evaluation metric so that it can be automated, benchmarked, learned, and analyzed. To this end, we formalize one type of exploratory analysis problem as a natural language generation task: goal **d**riven **d**iscovery of **d**ifferences between text **d**istributions via language **d**escriptions (D5). As shown in Figure 1, the input to the D5 task is a "problem" comprising a description of a user-specified exploration goal (understanding side effects) and a corpus pair (text samples from the distributions of self-reported reactions after taking each drug). The output is a "discovery" represented as a natural language predicate ("*mentions feelings of paranoia*"). We evaluate a discovery with two criteria (Section 3): (1) validity: it should describe a true difference (Zhong et al., 2022); and (2) relevance to the goal (McGarry, 2005).

---

[*]University of California, Berkeley, EECS Department. Email: ruiqi-zhong@berkeley.edu

37th Conference on Neural Information Processing Systems (NeurIPS 2023).

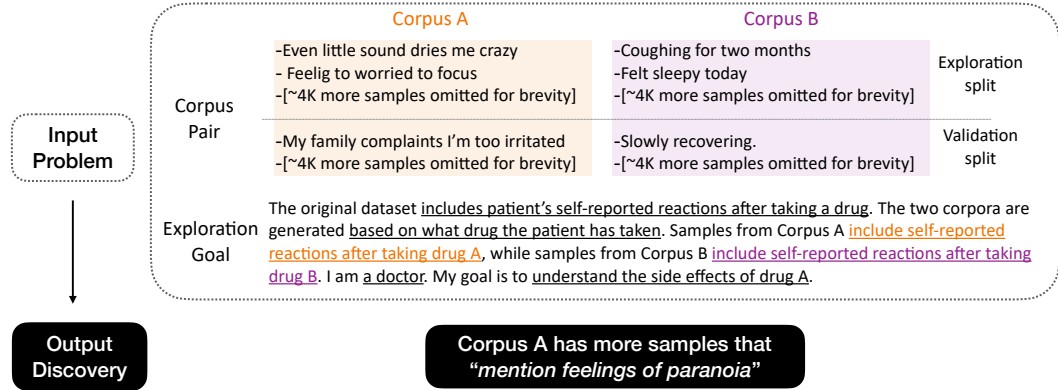

Figure 1: Each problem in OPEND5 contains 1) a corpus pair, which has ∼17K samples on average and is partitioned into two halves called "exploration split" and "validation split", and 2) a natural language description of the exploration goal, which also contains information about how the corpus pair was collected. A D5 system takes the goal and the exploration split as inputs and generates valid and relevant discoveries in natural language as outputs. The underlined texts in the exploration goal vary across problems, while the rest are templates.

Since D5 is open-ended and aims at discovering unknowns, the most popular benchmark practice—comparing system-generated outputs with human-written references on a test set—is infeasible. We therefore design two evaluation strategies.

- **Diagnostic**: we synthesized a dataset of D5 problems with known solutions, SYND5, to diagnose whether a D5 system can recover known differences between two synthetic corpora. This strategy is cheap and automated but might not reflect user utility in real applications.
- **Open-ended**: we collected a dataset, OPEND5, by aggregating 675 open-ended D5 problems ranging across business, social sciences, humanities, health, and machine learning (Figure 2), comprising 4.4 million text samples in total across problem corpora. We then manually evaluated a subset of the output discoveries. This strategy is subjective and expensive, but useful for obtaining qualitative insights on more realistic applications.

These two strategies allow us to quantitatively evaluate and compare D5 systems. For example, we compared 1) the system from Zhong et al. (2022) designed to describe corpus-level differences without goals, and 2) a goal-conditioned variant that we develop in Section 4. We found language models successfully use the specified goal: the goal-conditioned variant is correct 12% more often on SYND5, and it produces relevant candidate discoveries 31% more often on OPEND5.

We envision OPEND5 to be a growing, diverse repository of open-ended D5 problems. They will not only help us evaluate D5 systems more reliably, but also allow the following operations:

**Facilitate exploratory analysis.** Every time we build a better D5 system, we can apply it to a repository of open problems and send the discoveries to researchers who posed them. We show this paradigm is plausible by using our system to automatically produce useful discoveries on OPEND5 (Section 6.1), including insights from commercial reviews, temporal and demographic differences in discussion topics, political stances and stereotypes in speeches, differences in lyric styles, and error patterns in NLP systems. We anticipate future systems to produce more discoveries.

**Analyze the limitations of our evaluation.** Using concrete examples from OPEND5, we show that our current evaluation metrics do not encourage diverse findings, do not always produce causal conclusions, and cannot evaluate discoveries involving heavy expert knowledge (Section 6.2). More D5 problems can help us identify more limitations, which inform areas for future improvement.

**Train better D5 systems.** Like other ML tasks, we can train a system once we have a dataset. We describe a self-supervised learning algorithm that uses a repository of problems (without reference solutions) to train LMs to propose more valid hypotheses (Section 4.3). As a proof-of-concept, we show that it can make LMs better describe the differences between small groups of text samples.

To conclude, we show that D5 can be quantitatively evaluated, automated, analyzed, and learned. Like other ML tasks, it would benefit from a more diverse, authentic, and larger dataset. We hope future works can gather feedback from domain experts and curate an ever-larger dataset of D5 problems, thus accelerating exploratory analyses and facilitating scientific discoveries. [2]

## 2   Datasets: SYND5 and OPEND5

We first introduce how each input problem is formatted. Then we discuss 1) how we synthesized SYND5, which is used for automatic diagnostic evaluation, and 2) how we collected OPEND5, which is used to investigate the practical value of D5 systems in open-ended applications.

### 2.1   Task Format

Each D5 problem is represented by a corpus pair (Corpus A/B) and a description of the exploration goal. For example, Corpus A/B might be self-reported reactions after taking drug A/B, and the goal description would be "*comparing the side effects of drug A and drug B*". The desired output is valid and relevant discoveries in the form of natural language predicates (Figure 1), e.g. Corpus A has more samples that "*mentions feelings of paranoia*".

### 2.2   SYND5, a Diagnostic Benchmark with Reference Solutions and an Automatic Metric

To automatically diagnose a D5 system, we synthesized SYND5, a dataset of D5 problem with reference solutions. To synthesize Corpus A and Corpus B for each input problem, we used a language model (LM) to generate two corpora that simultaneously differ on two dimensions, one of which is goal-relevant and one of which is a distractor. For instance, suppose the goal is to "*understand how Corpus A differs from Corpus B in terms of topic*". Then we would synthesize an example where Corpus A is more sports-related while B is more art-related (goal-relevant: varying topic), while additionally Corpus A is in English while B is in French (distractor: varying language). The reference solution is the difference on the goal-relevant dimension, e.g. "*is sports-related*".

In more detail, to synthesize an example in SYND5, we first picked one goal-relevant and one distractor dimension from the set {topic, genre, language}, and sampled a value for each corpus and dimension (e.g. Corpus A: [sports, English]; Corpus B: [art, French]). We then synthesized Corpus A/B such that all its samples are in English/French (i.e. completely different on the distractor dimension) while $V$ percent of them are sports-related/art-related, where we varied $V$ from 0.6 to 1. Since the distractor difference is more salient, SYND5 penalizes D5 systems that ignore the goal and output the incorrect distractor difference "*is in English*". We synthesized 300 problems in total to create SYND5; see Appendix 9 for a detailed description of the pipeline.

To compute a D5 system's accuracy, we prompted `Claude-v1.3` (Bai et al., 2022b) to judge how often the output discovery is semantically equivalent to the reference. We construct the prompt by using 6 pairs of predicates with the labels of "equivalent", "similar", or "irrelevant" as few-shot examples, and ask `Claude-v1.3` to judge whether the output discovery and the reference are "equivalent". As a result, we can automatically diagnose a D5 system. See Appendix 12 for the prompt for equivalence judgement and Appendix 10 for two robustness checks, which (a) consider "similar" discoveries to be correct as well, and (b) use other LMs for equivalence judgement.

### 2.3   OPEND5, a Realistic Open-Ended Dataset without Reference Solutions

To evaluate a D5 system's utility under realistic applications, we also gathered OPEND5, a realistic dataset of 675 open-ended D5 problems. These problems range across business, social sciences, humanities, health, and machine learning; see Figure 2 for a few examples. To build OPEND5, two of the authors performed an extensive literature review on problems that could potentially benefit from our system, e.g., reading survey papers (Nguyen et al., 2020) and courses on computational social sciences, and skimming through the ACL proceedings from the past decade and datasets from Kaggle that have an NLP tag; we then annotated the exploration goals, scraped/generated the

---

[2]Our code is released at `https://github.com/ruiqi-zhong/D5` and our code to download OPEND5 is released at `https://github.com/petezh/OpenD5`. Given the limitations of our system, practitioners should interpret its outputs with caution and not use it to fully automate scientific discoveries.

| Domain | Example Datasets | How the Corpus Pairs are Generated | |
|---|---|---|---|
| | | Corpus A | Corpus B |
| | | **87 Business problems** | |
| Commercial Reviews | Airline reviews | 1st-class passenger reviews | Economy passenger reviews |
| | Product Reviews | Reviews that give 10 stars | Reviews that give 0 star |
| Finance | YC startups | Successful startup descriptions | Failed startup descriptions |
| | News Headlines | Top headlines when S&P rises | Top headlines when S&P falls |
| | | **278 Social Sciences problems** | |
| Politics | Administration policy | Admin policy from Trump | Admin policy from Obama |
| News | Reuters headlines | Headlines from 2014 | Headlines from 2015 |
| Language | Craiglist Negotiations | Dialogue from successes | Dialogue from failures |
| | Diplomacy Dialogues | Lies | Honest statements |
| Sociology | Happy moments | Self-reported happy moments from females | Self-reported happy moments from males |
| | Rate My Professor | Reviews of female lecturers | Reviews of male lecturers |
| | | **169 Humanities problems** | |
| Arts | Music lyrics | Drake rap lyrics | Kanye rap lyrics |
| Education | Student essays | Essays that received full score | Essays with only partial credit |
| | | **10 Health problems** | |
| Health | Doctor's note | Patients diagnosed with pneumonia | Patients not diagnosed with pneumonia |
| | | **131 Machine Learning problems** | |
| Machine Learning | NLI — distribution shift | Samples from SNLI | Samples from MNLI |
| | QQP — spurious correlation | Individual questions with label "paraphrase" | Individual questions with label "non-paraphrase" |
| | LM's output | Generations from one LM | Generations from another LM |
| | inputs — error analysis | Inputs where one model is correct | Inputs where one model is wrong |
| | WikiText — clustering | Samples from one cluster | Samples not from a cluster |

Figure 2: OPEND5 contains 675 problems. See citations in Appendix 23.

corresponding corpora, and post-processed them over nine months (see complete list of citations in Appendix 23). As shown in Figure 1, each goal describes the original dataset, how the two given corpora are generated, who is using the system, and what property of the two corpora does the user want to understand. Each OPEND5 problem is reviewed by at least two of the authors to reduce grammatical mistakes and ambiguous interpretations of the goal.

Each corpus contains around 17K text samples on average, and OPEND5 in total comprises 4.4 million distinct text samples. We use 50% of each corpus as the "exploration" split and 50% as the "validation" split. The system can only access the exploration split, while the validation split is reserved for the evaluators to validate the discovery. A validation split prevents overfitting the discoveries to the given samples and is analogous to the train-test split in machine learning.

Since we hope to build systems that can tackle challenging open-ended problems, we did not avoid cases where we do not know the ground truth answer. This is different from standard benchmarking practices, where humans can provide a reference solution to evaluate an AI system. However, even though we do not know the ground truth, once a system produces a discovery, we can still evaluate it. We present our evaluation metrics in the next section.

## 3    Evaluation Metrics for Open-Ended D5 problems

For the goal of comparing the side effects of drug A and drug B, how do we evaluate a system-generated discovery that Corpus A "*mention feelings of paranoia*" more often? First, it needs to be valid, such that indeed more samples from Corpus A satisfy this predicate, which can be evaluated (approximately) objectively. Second, it needs to be relevant to the goal of understanding side effects, which depends on the user's subjective judgement. We define validity and relevance below.

**Validity.** Similar to Zhong et al. (2022), we require an output discovery $h$ to be a truth predicate on a text sample. For example, if $h$ = "*mentions about family*", then $h$ is true on the string $x_1$ = "*My daughter loves me*" and false on the string $x_2$ = "*I'm going to school*". Define $T(h, x) \in [0, 1]$ as "the certainty that $h$ is true on $x$", e.g., $T(h, x_1) \approx 1$ and $T(h, x_2) \approx 0$. We approximate $T(h, x)$ by asking three Turkers how certain they are and averaging their responses (see Appendix 11 for details).

Let $\mathcal{D}_A^{\mathrm{val}}$ and $\mathcal{D}_B^{\mathrm{val}}$ denote the validation sets for Corpus A and B. We define the "validity" $V$ as

$$V(h) := \mathbb{E}_{x \sim \mathcal{D}_A^{\mathrm{val}}}[T(h, x)] - \mathbb{E}_{x \sim \mathcal{D}_B^{\mathrm{val}}}[T(h, x)]. \tag{1}$$

Computing $V(h)$ is expensive since it requires human annotations $T(h, x)$ on a set of text samples even to evaluate a single discovery $h$. In practice, we do not have the budget to compute $V(h)$ on the entire validation split; therefore, we approximate this quantity by randomly sampling from Corpus $A$ and Corpus $B$. We use these samples to compute an empirical estimate of $V$, as well as a $p$-value for the null hypothesis that $V \leq 0$ using a one-sided t-test.

**Relevance.** A discovery may be irrelevant even if $V = 1$. For example, if the goal is to understand the writing style differences between higher-scoring essays (Corpus A) and lower-scoring ones (Corpus B), the discovery that Corpus A "*achieves higher scores*" has high validity score by definition but irrelevant to the goal of understanding stylistic differences.

Therefore, we designed a procedure to evaluate relevance, where human or language model evaluators would score each discovery with ②/①/⓪. The evaluators used the rubric below, which illustrates the meaning of each score with the essay example above:

- ②, relevant; e.g. the discovery "*write in first person*" is directly related to the writing style.
- ①, indirectly relevant; e.g. the discovery "*use the word "I"*", is not exactly a writing style, but can still inform the relevant underlying principle of "*write in first person*".
- ⓪, irrelevant; e.g. the discovery "*argue for abortion*" is unrelated to the writing style.

To minimize biases while comparing two systems, the evaluators are blind to which system generates which discoveries.

To conclude, an ideal discovery would have a high $V$ value with a small $p$-value and achieve ratings of ② in relevance. In the next section, we will build a D5 system that addresses these criteria by first proposing goal-relevant candidate discoveries (hypotheses) and then automatically validate them.

**Other metrics.** We also explored two other subjective metrics, novelty (how difficult it is to generate the discovery) and significance (how beneficial it is to learn about the discovery). Due to space limit, we present their rubrics and related results in Appendix 14.

## 4 Methods: Building a D5 System

We describe our D5 system, which maps from a corpus pair and an exploration goal to a set of natural language predicates. Our system is inspired by a two-stage model of how humans discover patterns in data: creatively brainstorming hypotheses and then rigorously validating them on the data (Ludwig & Mullainathan, 2022). Analogously, we first propose hypotheses conditioned on the exploration goal and a subset of samples from the corpus pair (Section 4.1). We then use a language model to approximately compute the validity of each hypothesis, and output the valid ones as the final discoveries (Section 4.2). Our system closely mirrors that of Zhong et al. (2022), except that we leverage the goal to propose more relevant hypotheses. Finally, we present a self-supervised learning algorithm to improve an LM's ability to propose more valid hypotheses (Section 4.3); however, due to API access constraint, we cannot apply it to fine-tune `gpt-3`, so we provide a proof of concept experiment on `Flan-T5` (Chung et al., 2022).

### 4.1 Hypothesis Proposer

We prompt `gpt-3` (Ouyang et al., 2022) to propose hypotheses. Denoting the exploration split of Corpus A/B as $\mathcal{D}_A^{\text{exp}}/\mathcal{D}_B^{\text{exp}}$, we construct the prompt by concatenating a few random samples from $\mathcal{D}_A^{\text{exp}}$ and $\mathcal{D}_B^{\text{exp}}$, the exploration goal, and an instruction to output a list of hypotheses. Figure 3 (left) depicts an example of the resulting prompt, together with a typical language model output.

Since the entire corpus pair might not fit into one prompt, we construct multiple prompts with different sets of samples so that `gpt-3` can "see" as many different samples as possible in our pipeline. We continue sampling hypotheses with different prompts until obtaining a set of 60 hypotheses, which we call $H_{\text{init}}$. Appendix 15 includes more details on selecting the sets of samples for different prompts.

### 4.2 Hypothesis Validator

Many hypotheses in $H_{\text{init}}$ have low validity: they are not more often true on $\mathcal{D}_A$ than on $\mathcal{D}_B$ (i.e. $V(h) \leq 0$). To automatically filter them out, we use a language model $T'$ to simulate the Turkers'

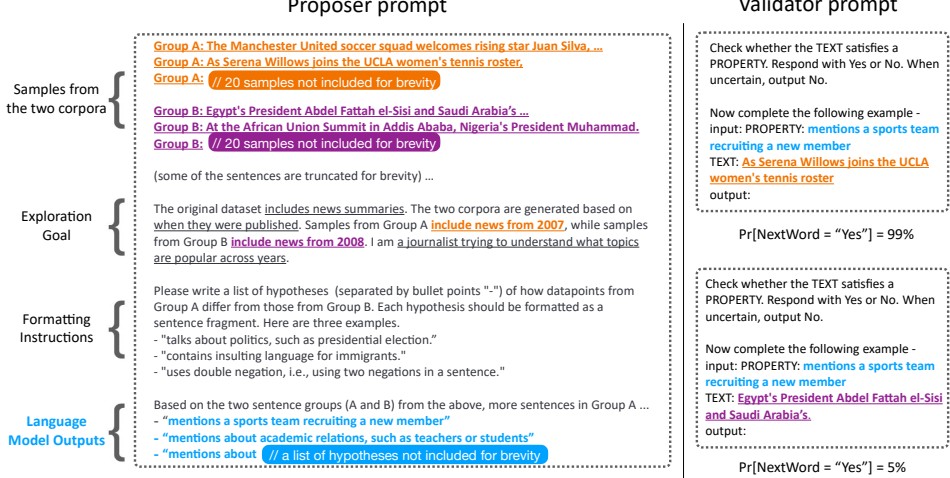

Figure 3: All underlined content in the prompt differs across problems, while the other content in the prompt is templated. **Left**: proposer prompt. The generated hypotheses are in blue. All content with colored background is excluded for brevity. For the baseline of not using the exploration goal, we removed the "exploration goal" block from the prompt. **Right**: the validator prompt.

judgement $T$ and hence approximate the validity score $V(h)$ with the function $V'(h)$, defined as

$$V'(h) := \mathbb{E}_{x \sim \mathcal{D}_A^{\exp}}[T'(h, x)] - \mathbb{E}_{x \sim \mathcal{D}_B^{\exp}}[T'(h, x)]. \tag{2}$$

To compute $T'$, we ask `Flan-T5` whether $x$ satisfies $h$ with the prompt shown in Figure 3 (right). To better simulate Turker's judgment, we collected additional Turker annotations to fine-tune `FLAN-T5` (see Appendix 16 for details about the data collection process). We then obtain a significance value $p'$ by performing a t-test to compare the mean value of $V'(h, x)$ on the exploration split of Corpus $A$ to that of Corpus $B$, rule out the hypotheses with $p'$ greater than 0.001, and output the remainder as discoveries. Finally, we obtain additional discoveries by repeating the same process but asking our system to propose and validate hypotheses about Corpus $B$ rather than Corpus $A$. Appendix Figure 5 visualizes our entire pipeline and Appendix 8 discusses the computational resources we used.

### 4.3 Self-Supervised Learning with Open-Ended Problems: A Proof of Concept

Since D5 problems are open-ended, future systems could potentially produce discoveries with higher validity scores than any known discovery. Therefore, we design a self-supervised learning algorithm to improve an LM's ability to propose more valid hypotheses, using the principle that **it is easier to validate a discovery than to generate one**.

**Algorithm.** Suppose we are given a set of problems for training and an initial language model $m_0$. Our goal is to automatically generate a set of *prompt-completion* pairs to fine-tune $m_0$ so that it can propose hypotheses that are more valid. To generate a *prompt*, we randomly sample a problem and create a proposer prompt following the procedure in Section 4.1. To generate the desired *completion* given a prompt, we sample multiple hypotheses from $m_{\text{init}}$, approximate their $V'$ score on the samples in the proposer prompt with the same language model $m_{\text{init}}$ (Section 4.2), and select the highest scoring hypothesis. Finally, we use the prompt-completion pairs to fine-tune $m_0$.

**A Proof of Concept Experiment.** Since we cannot fine-tune `text-davinci-003`, we can only experiment with `Flan-T5-xxl` (Chung et al., 2022), an open-sourced instruction-tuned model that might only work well for easier "mini-problems". As a proof of concept, we tested the above self-supervised learning algorithm on the task of describing groups of four samples, where each group comes from a text cluster.

We computed both the automated "self-evaluation" validity score $V'$ and the "true" validity score $V$ according to Turker evaluation for evaluation. After self-training, $V'$ improves substantially from 0.22 to 0.37, and the $V$ improves from 0.07 to 0.10, with a $p$-value of 0.02. This result provides preliminary evidence that self-training could be applied to a large set of problems to improve the

| text-davinci-003 | w/ goal | wo/ goal | gpt-4 | w/ goal | wo/ goal |
|---|---|---|---|---|---|
| w/ validator | 12% | 2% | w/ validator | 27% | 15% |
| wo/ validator | 4% | 1% | wo/ validator | 8% | 5% |

Table 1: The accuracy on SYND5 using different proposers, with/without incorporating goals, and with/without using validators. Using the validator, the goals, and `gpt-4` leads to better results.

| Hypothesis Relevance | ② | ① | ⓪ | average |
|---|---|---|---|---|
| Using the goal | 79% | 9% | 12% | 1.68 |
| Not using the goal | 52% | 16% | 32% | 1.20 |

Table 2: How often the hypotheses proposed by `text-davinci-003` are rated by the authors as ②/①/⓪ in terms of relevance (Section 3). Overall, using the goal significantly increases relevance.

validity of the hypotheses; we expect future validators to simulate human judgments better, hence decreasing the approximated gap of improvement between $V$ and $V'$. We discuss more training and evaluation detail in Appendix 20.

# 5 Quantitative Evaluation on SYND5 and OPEND5

We show that both SYND5 and OPEND5 can be used to quantitatively evaluate D5 systems. Since SYND5 is automatic, we used it to compare a broad range of D5 systems and studied the contributions of three different factors: the quality of the proposer model (`gpt-4` vs. `text-davinci-003`), the use of a validator, and the use of a goal. We then further investigated the effect of using goals under realistic applications through human evaluation on OPEND5.

**Automatically comparing different variants with SYND5.** As mentioned above, we ablated 3 factors, resulting in $2^3 = 8$ variants. We compared 1) using `text-davinci-003` vs. `gpt-4` as the hypothesis proposer; 2) using the validator to compute $V'$ for each hypothesis and outputting the highest-scoring hypothesis, vs. not using the validator and outputting a random hypothesis; and 3) using the goal vs. replacing it with "*I want to understand how Corpus A is different from Corpus B.*". We then automatically calculated the accuracy for each variant as described in Section 2.2.

We report the results in Table 1. We find that using the validator and the goals significantly improve the performance, and `gpt-4` outperforms `text-davinci-003` with goals and the validator ($p < 1\%$ under a t-test). We conducted two additional robustness checks in the Appendix 10: (a) using `text-davinci-003` instead of `Claude-v1.3` to judge predicate equivalence, and (b) considering discoveries semantically similar to the references also to be correct; our conclusions do not change.

Finally, to improve the accessibility of our research, we ran the same experiments using `gpt-3.5-turbo` and `flan-t5-xxl` as our proposer, and report the results in Appendix Table 6. To show that our conclusions are general and not only apply to synthetically generated texts, we additionally constructed an extension of SYND5 with human-written texts by adapting the NYT dataset from Wang et al. (2023), where each text sample is a New York Times article with a topic and a location label: the topic dimension has 9 different values (e.g., politics, arts) and the location dimension has 10 different values (French, Italy); we then followed the same procedure described in Section 2.2 to create this extension of SYND5, and report our systems' performance in Appendix Table 7. In all experiments, using the validator and the goal improves the performance.

**Investigating whether using goals improves relevance on OPEND5.** We then investigated whether `text-davinci-003` can leverage the goals to propose more relevant hypotheses on more realistic applications in OPEND5. We sampled 100 problems from OPEND5 with distinct goals and randomly sampled 2 hypotheses from `text-davinci-003` with/without using goals (see Figure 3), resulting in 400 hypotheses to evaluate. Three authors then rated their relevance based on the rubric in Section 3, while being blinded about which hypotheses were generated with the goal. Our main paper focuses on presenting the evaluations performed by ourselves, since crowdworkers might be noisy and untrustworthy (Veselovsky et al., 2023; Suhr et al., 2021).

We report the results in Table 5. Since this evaluation is subjective, the inter-annotator agreement is only moderate (Kappa=0.56); however, we can still robustly conclude that `text-davinci-003` can leverage goals to propose hypotheses with higher average relevance rating, since this conclusion can

be independently reproduced by every individual evaluator with $p < 10^{-8}$. To make sure that the same conclusion can be robustly reproduced by external non-authors, we also evaluated the relevance of the hypotheses with Amazon Mechanical Turks, `gpt-3.5-turbo`, `Claude-v1.3`, and `gpt-4`. We report the results in Appendix Table 8 and found that our conclusion robustly holds under five different types of evaluators, including expert authors, external crowdworkers, and language models from different companies with different levels of capabilities.

Finally, we conducted similar experiments for the novelty and significance metrics in Appendix 14 and found that they both benefit from using goals as well. In the next section, we present example discoveries on OPEND5 to qualitatively understand what a D5 system can achieve.

# 6 Qualitatively Analyzing Discoveries and Limitations with OPEND5

To understand the utility and the limitation of a D5 system, we ran it on OPEND5, a set of realistic D5 problems, and analyze the output discoveries qualitatively.

## 6.1 Producing Discoveries on OPEND5 and Analyzing Them

We ran our D5 system on OPEND5, producing 3296 discoveries in total. However, we do not have enough budget to validate every finding, since estimating $V$ is expensive (Section 3). Therefore, from the remaining 3296 discoveries, we manually selected 21 discoveries that 1) achieve a relevance score of ②, 2) are representative of potential use cases, 3) do not require expert knowledge for Turkers to judge, and 4) are likely to achieve a small $p$-value with fewer than 200 samples from $\mathcal{D}^{\text{val}}$ .

We then estimated their validity based on the procedure described in Section 3 by using fewer than 200 samples from the validation split and calculated the $p$-values, which cost us ∼$1500 in total on MTurk. Since we are testing multiple discoveries and each of them can be statistically significant merely due to chance, we keep 13 discoveries with $V$ that are significantly non-zero with $p$-value below 7%, a threshold determined by the Benjamini Hochberg's procedure with a false discovery rate of 10%. In other words, <10% of the discoveries presented are false discoveries in expectation.

We detail 5 of the 13 discoveries in this section, with the remainder in Appendix 18. For each discovery, we report its automated validity score $V'$, the estimated true validity score $V$, and their respective $p$ values in Table 3.

**Understanding political stances and stereotypes in speeches.** When comparing presidential speeches on immigrants from Obama to those from Trump, the former "*argues for a path forward to promote the fair and just treatment of immigrants*", while the latter more frequently "*refers to illegal immigrants as criminals*".

**Analyzing errors in NLP systems.** We fine-tuned a pair of models on two different natural language inference datasets, (a) MNLI and (b) SNLI. To understand their patterns of errors, we defined Corpus A to be the subset of MNLI where a is right and b is wrong, and Corpus B to be where b is right and a is wrong. We found that the latter more often "*has an informal tone, such as slang or colloquial speech*". One possible explanation is that MNLI contains more different genres and hence more informal speeches, causing the former model to perform better on these examples.

| Output discovery | $V$ | $p$ | $V'$ | $p'$ |
|---|---|---|---|---|
| "*argues for a path forward to promote the fair ...*" | 0.16 | 1.26e-04 | 0.35 | 2.01e-73 |
| "*refers to illegal immigrants as criminals*" | 0.09 | 6.17e-03 | 0.19 | 3.17e-38 |
| "*has an informal tone, such as slang or colloqu...*" | 0.08 | 2.35e-03 | 0.24 | 1.46e-35 |
| "*mentions lack of legroom* " | 0.16 | 1.15e-03 | 0.38 | 1.34e-45 |
| "*mentions children or family*" | 0.08 | 1.00e-05 | 0.11 | 8.05e-09 |

Table 3: A subset of discoveries presented in Section 6.1 and their associated estimated validity score $V$, validity score approximated by a model $V'$, and their respective $p$-values $p$ ($p'$) for the null hypothesis that $V(V') < 0$ under a t-test. We present the full set of 13 discoveries in Table 10.

**Analyzing airline customer reviews.** We compared the concerns in reviews of the airline Air Canada v.s. its subsidiary, Air Canada Rogue, which is considered a low-price wing of Air Canada. The latter more often "*mentions lack of legroom*".

**Analyzing gender differences in self-reported happy moments.** Compared to self-reported happy moments written by males, those by females "*mentions children or family*" more often. Caution: misinterpreting this correlation as causation could reinforce societal biases (Section 6.2).

Due to space constraints, we list more examples on analyzing distribution shifts, text clusters, lyric styles, and news headlines in Appendix 18 and their associated $V$ and $V'$ values in Appendix Table 10. Across these discoveries, the approximated validity score $V'$ has a 71% spearman rank correlation with human rating $V$ (66% for Pearson correlation), thus providing informative yet unreliable signals to practitioners about their validity. We hope that $V'$ can better approximate $V$ values in the future as the quality of the validators improve. Finally, future works can collect more open problems, allowing D5 systems to produce more impactful discoveries.

### 6.2 Concrete Examples in OPEND5 Inform Limitations of D5 Evaluation

We discuss limitations of D5 evaluation in this section using concrete examples from OPEND5.

**Our metrics do not evaluate diversity.** There are often multiple valid and relevant discoveries, and our system ideally should generate all of them. For example, when comparing low-rating and high-rating reviews to understand what stands out to customers, both "*mentions the hidden fees and poor customer service at the airport*" and "*mentions the airline charging extra for carry-on items*" could be valid discoveries. Our current evaluation does not reward diverse discoveries, and the current system sometimes repeats a discovery using similar paraphrases, e.g., "*mentions the rude and unprofessional attitude of the staff*" and "*mentions the staff being rude and unhelpful*". Future evaluation metrics can take diversity into account.

**Interpreting discoveries requires domain experts.** We used Turkers' judgment when computing $T(h, x)$ to judge the validity of a discovery. However, many discoveries require expert knowledge to interpret properly. For example, it requires medical training to reliably judge whether a self-reported drug-use experience satisfies "*mentions psychedelics, such as LSD and shrooms.*"

**Correlation $\neq$ causation.** Our metrics currently do not evaluate whether the discovery is causally related to how the corpus pair was generated. For example, when comparing self-reported happy moments from females and males, even if the former corpus has more samples that "*mention children and family*", it does not necessarily imply family plays a more important role in inter-personal relations for females; an alternative hypothesis is that females might mention people in general more often than males do, hence leading to the observation that they mention family more often. Spurious correlations could also sneak into our validity evaluation: for example, if the Turkers implicitly associate female activities as family-related Greenwald & Banaji (1995), then we might falsely make this discovery due to evaluator biases. Future metrics should also consider plausible alternative hypotheses to evaluate causality and control the potential biases from the human evaluators. Additionally, we should treat the discovery from D5 with caution to prevent automating and amplifying societal biases.

We discuss other limitations, such as restricting the discovery to be a single predicate, the biases in authors' qualitative evaluation, and the incomprehensiveness of OPEND5 in Appendix 19.

## 7 Related Work and Discussion

**Inductive Reasoning with NLP Models.** Recent works show that language models are capable of inductive reasoning under restricted settings, discovering patterns from a set of text data points and describing them with language (Honovich et al., 2022). Yang et al. (2022) use this capability to induce natural language rules with the format of "*if . . . then . . .*". Zhou et al. (2022) and Ye et al. (2022) use this capability to improve zero/few-shot accuracy by inferring the most likely instruction using input-output example(s) of the target task. Zhong et al. (2022) and Singh et al. (2022) use this capability to discover patterns in datasets, and we improve by building an automatic benchmark and a dataset of open-ended problems and require the discovery to be relevant.

ML models can also perform inductive reasoning in other modalities, such as vision. Hernandez et al. (2021) describes visual features that activate a neuron; Zhu et al. (2022) describes distribution

shifts between the training distribution and the test distribution for images; and Eyuboglu et al. (2022) describes errors made by vision models. We hope future models can perform inductive reasoning in other modalities, such as sound (Aghajanyan et al., 2023) or physical senses (Thomason et al., 2016).

**Exploratory Analysis and Automated Discovery.** It is not new to automatically discover patterns by learning from empirical data. To list a few classical methods, linear regression analyzes the effect of each real-valued feature by interpreting the learned weights (Draper & Smith, 1998); n-gram models can extract discriminative phrases, thus yielding insights about corpus-level differences (Manning & Schutze, 1999); topic models (Blei et al., 2003) can extract major topical variations across documents, where each topic is represented as a distribution over words; small decision trees can extract interpretable if-then statements (Letham et al., 2015); and an entity embedding model learned on existing relations between entities can predict unseen relations (Socher et al., 2013). In comparison, D5 produces discoveries in the form of natural language predicates, which are interpretable and can express abstract concepts; additionally, it is more directed at the goal, while machine learning classifiers like naïve bayes or linear regression will pick up any discriminative features: Appendix 21 offers a more comprehensive discussion using examples from SYND5. Given the respective strength of D5 and traditional exploratory methods, we envision D5 to serve as a complementary method to traditional methods.

**Epistemology.** While the process of validating a hypothesis is well-formulated, it is much less well-understood how to automatically generate hypotheses and decide what discoveries are meaningful (Shapere, 1964; Heckman & Singer, 2017). Related works in this area have been sparse, among which McGarry (2005) sketches high-level principles for evaluating knowledge discoveries and Ludwig & Mullainathan (2022) proposes to crowd-source hypotheses from MTurk workers. We concur with the perspective of Polanyi et al. (2000) that meaningfulness of a hypothesis cannot be explicitly verbalized with simple logic but is dependent on implicit community norms; therefore, the process of proposing hypotheses should be learned from empirical data (e.g. pre-training, self-training, or human feedback) rather than deduced from a priori analysis of concepts (Quine, 1969). We hope contributions from other domains can provide more empirical data on what discoveries are meaningful, hence guiding our system to produce more important discoveries.

# Acknowledgement

We thank Xiaochuang Han and Sam Bowman for their early discussions on this project. We thank Cathy Chen, Erik Jones, Jessy Lin, Alex Pan, Chenglei Si, Xi Ye, and Tianyi Zhang for their helpful feedback on the paper draft. We thank OpenAI and Anthropic for providing model access.

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

# 8 Cost for Running the Experiments

For each problem, we ran the proposer for 10 times on average; assuming each prompt to be at most 4000 tokens, we spent around $2.4 for each problem on `OpenAI` APIs if we use `gpt-4` and `text-davinci-003`, and the cost would decrease to $0.8 if we use `gpt-3.5-turbo`. Notice that these estimates are computed based on the prices as of 05/14/2023, and we expect the price to further decrease in the future. We ran the Flan-T5 based validator for $\tilde{2}$ hours on 1 80G A100 GPUs.

The total amount of computational resources spent in this research paper is around $2,500 in terms of `OpenAI` API and 3,000 hours of compute on A100 GPU with 80G memory.

# 9 Generation Process of SYND5

The high-level description is in Section 2.2. Here we discuss the procedure that generated SYND5.

We consider three dimensions of differences: topic, genre, and style. For each, we generated 14/9/7 values, e.g., "*celebrity love stories*" and "*sports team recruiting athletes* for the topic attribute, "*rap lyrics*" and "*screen play*" for the style attribute, and "*French*" and "*Spanish*" for the language attribute. We then used GPT-4 and the Claude API to synthesize 54K text samples, where for each text sample we sampled a topic, genre, and style randomly, e.g. "*Write a rap about a sports team recruiting athletes in French*". To synthesize a random SYND5 problem, we randomly sampled a distractor dimension (e.g. language) and a target dimension (e.g. topic), and for each dimension we sampled two random values (e.g. English and French for language, sports and art for topic).

For each problem, we sampled 10 texts for corpus $A$ such that all of them satisfy one sampled value for the distractor dimension (e.g. corpus $A$ is entirely in English), and 10 texts for corpus $B$ for to satsify the other distractor dimension (e.g. corpus $B$ is entirely in French). Then we set $V$ fraction of corpus $A$ to satisfy the reference target attribute, e.g. "is sports-related", and $f$ fraction of corpus $B$ to satisfy the other value for the target dimension (e.g. "is art-related"). We chose $V$ uniformly at random from [0.6, 0.8, 1]. Finally, we provide $k$ example hypotheses from the target dimension other than the target dimension values for Corpus A and Corpus B, and we chose $k$ from [0, 2] uniformly at random. We then sampled 300 D5 problems in total from this distribution.

# 10 Robustness Checks for Results on SYND5

Table 4 shows the accuracy of different systems using `text-davinci-003` as the judge for semantic equivalence. Table 5 shows the accuracy of different systems if we consider outputs semantically similar to the reference to be correct. Across all setups, we found that the conclusion reached in Section 5 still holds under these robustness checks.

| `text-davinci-003` | w/ goal | wo/ goal | `gpt-4` | w/ goal | wo/ goal |
|---|---|---|---|---|---|
| w/ validator | 6% | 1% | w/ validator | 23% | 9% |
| wo/ validator | 3% | 0% | wo/ validator | 6% | 2% |

Table 4: Same Table as 1, except that we use `text-davinc-003` instead `Claude-v1.3` to judge similarity. Using the validator, the goals, and `gpt-4` leads to better results.

| `text-davinci-003` | w/ goal | wo/ goal | `gpt-4` | w/ goal | wo/ goal |
|---|---|---|---|---|---|
| w/ validator | 46% | 23% | w/ validator | 53% | 43% |
| wo/ validator | 24% | 16% | wo/ validator | 24% | 24% |

Table 5: Same Table as 1, except that we calculate how often the output is similar, rather than equivalent, to the reference. Using the validator, the goals, and `gpt-4` leads to better results.

To improve the accessibility of our research, we ran the same experiments with `gpt-3.5-turbo` and `flan-t5-xxl`, and report the results in Appendix Table 6. To show that our conclusions are general and not only apply to synthetically generated texts, we additionally constructed an extension of SYND5 with human-written texts by adapting the NYT dataset from Wang et al. (2023), where each text sample is a New York Times article with a topic and a location label: the topic dimension

has 9 different values (e.g., politics, arts) and the location dimension has 10 different values (French, Italy); we then followed the same procedure described in Section 2.2 to create this extension of SYND5, and report our systems' performance in Appendix Table 7. Under all experimental setups, using the validator and the goal improves the performance.

|  | w/ g, w/ v | wo/ g, w/ v | wo/ g, w/v | wo/ g, wo /v |
|---|---|---|---|---|
| flan-t5-xxl | 0.05 | 0.03 | 0.02 | 0.01 |
| gpt-3.5-turbo | 0.27 | 0.10 | 0.08 | 0.03 |
| gpt-4 | 0.27 | 0.15 | 0.08 | 0.05 |

Table 6: Similar to Table 9, we used `gpt-3.5-turbo` and `flan-t5-xxl` as the proposer to tackle the SynD5 dataset, and report the performance with/without using the goal (g), and with/without using the vadliator (v). We find that using the goal and the validator significantly improves the performance, and open-sourced models lag significantly behind. Additionally, `gpt-4` does not significantly outperform `gpt-3.5-turbo`.

|  | w/ g, w/ v | wo/ g, w/ v | wo/ g, w/v | wo/ g, wo /v |
|---|---|---|---|---|
| gpt-3.5-turbo | 0.61 | 0.24 | 0.22 | 0.10 |
| gpt-4 | 0.55 | 0.28 | 0.22 | 0.16 |

Table 7: We created an extension of SynD5 by adapting a dataset of New York Times articles with two dimensions: topic and locations, each with 9 and 10 values. We then used `gpt-3.5-turbo` and `gpt-4` as the proposer, and found the same conclusion: using the goal and a validator improves the performance. Additionally, `gpt-4` does not significantly outperform `gpt-3.5-turbo`.

# 11 Computing Turker Judgement

**Scoring.** To estimate $T(h, x)$ with Turker's rating, where $h$ is a truth predicate of a text sample $x$, the Turker needs to read $h$ and $x$ and then choose among six options: "Certainly Yes", "Likely Yes", "Neutral", "Likely No", "Certainly No", and "Confusing/Cannot be decided." For each $(h, x)$ pair, we collect responses from three Turkers. To compute the average across them, we collect a list of scores using the following rule: each "Certainly Yes" would receive a score of 1.00, "Likely Yes" 0.75, "Neutral" 0.50, "Likely No" 0.25, "Certainly No" 0.00, and "Confusing/Cannot be decided." receive two scores of 0.50. We then take the average over all the scores we collected from the Turkers for one $h$ and $x$. "Confusing/Cannot be decided." receives two scores of 0.50 because we want such a response to drag the average rating towards neutral and it has a larger effect than choosing "Neutral".

**Payment.** We adjust the payment for each HIT task based on the number of words they need to read. We pay them approximately 0.001 cent per word, and using the conservative estimate that adults read about 200 words per minute, we pay them around $12 per hour. We spent in total around $5K on this HIT task.

**Qualification.** We only recruited Turkers who are located in the U.S. Additionally, we designed qualification test with 8 questions; the questions are designed to be easy to answer as long as they have read our instructions below, and we only accepted turkers who made mistakes on at most one questions.

**Annotation Instruction.** We show our annotation instruction below. We only show examples of choosing "Certainly Yes", "Certainly No", and "Confusing" to encourage the Turkers not to choose neutral ratings. Additionally, we explicitly tried to address Halo effect – where the text does not satisfy a predicate $h$ but satisfies a predicate $h'$ that is highly correlated with $h$. For example, for the text sample $x$ = "*Really love the flight!!*" does not satisfy the predicate $h$ = "*mentions that the breakfast is good on the plane*", even though it satisfies a highly correlated predicate $h'$ = "*likes the flight.*"

## 11.1 Instructions

Below are the same instructions we have shown you during the qualification. Thanks for visiting this page and refresh your memory about the instruction!

**Instruction**: In this task, you will check whether a TEXT satisfies a PROPERTY

**Example 1**
**Property**: mentions a natural scene.
**Text**: I love the way the sun sets in the evening.

- A) Certainly Yes.
- B) Likely Yes.
- C) Neutral.
- D) Likely No.
- E) Certainly No.
- F) Confusing/Cannot be decided.

**Answer.** A. sun set is nature-related; if you feel a bit ambivalent, B is also acceptable.

**Example 2**
**Property**: writes in a 1st person perspective.
**Text**: Makima is cute.

- A) Certainly Yes.
- B) Likely Yes.
- C) Neutral.
- D) Likely No.
- E) Certainly No.
- F) Confusing/Cannot be decided.

**Answer.** E. This text is undoubtedly written in the 3rd person perspetive, so E.

**Example 3**
**Property**: is better than group B.
**Text**: I also need to buy a chair.

- A) Certainly Yes.
- B) Likely Yes.
- C) Neutral.
- D) Likely No.
- E) Certainly No.
- F) Confusing/Cannot be decided.

**Answer.** F. It is unclear what the hypothesis mean (e.g., what does group B mean?) and doesn't seem related to the text. So F.

**Example 4**
**Property**: mentions that the breakfast is good on the airline.
**Text**: The airline staff was really nice! Enjoyable flight.

- A) Certainly Yes.
- B) Likely Yes.
- C) Neutral.
- D) Likely No.
- E) Certainly No.
- F) Confusing/Cannot be decided.

**Answer.** E. Although the text appreciates the flight experience, it DOES NOT mention about the breakfast. So the answer is E.

**Example 5**
**Property**: appreciates the writing style of the author.
**Text**: The paper absolutely sucks because its underlying logic is wrong. However, the presentation of the paper is clear and the use of language is really impressive.

- A) Certainly Yes.
- B) Likely Yes.
- C) Neutral.
- D) Likely No.
- E) Certainly No.
- F) Confusing/Cannot be decided.

**Answer.** A. Although the text dislikes the paper, it DOES like the writing style. So the answer is A.

## 12   Prompt to Judge Predicate Similarity

We prompt Claude v1.3 (Bai et al., 2022b) to judge whether the predicated predicate is similar to the reference. We consider a response that leads to a "yes" to be correct when we require the discovery to be semantically equivalent to the reference, and consider a response that leads to a "yes" or "related" to be correct when we require the discovery to be semantically similar to the reference.

" *Is text_a and text_b similar in meaning? respond with yes, related, or no.*

*Here are a few examples.*
*Example 1:*
*text_a: has a topic of protecting the environment*
*text_b: has a topic of environmental protection*
*and sustainability*
*output: yes*

*Example 2:*
*text_a: has a language of German*
*text_b: has a language of Deutsch*
*output: yes*

*Example 3:*
*text_a: has a topic of the relation between political figures*
*text_b: has a topic of international diplomacy*
*output: related*

*Example 4:*
*text_a: has a topic of the sports*
*text_b: has a topic of sports team recruiting new members*
*output: related*

*Example 5:*
*text_a: has a named language of Korean*
*text_b: uses archaic and poetic diction*
*output: no*

*Example 6:*
*text_a: has a named language of Korean*
*text_b: has a named language of Japanese*
*output: no*

*Target:*
*text_a: {**predicate**}*
*text_b: {**reference**}*
*output:*"

# 13   Relevance Rating with External Non-Authors

To make sure that the conclusion that "using goal in the context can improve hypotheses relevance" can be robustly reproduced by external non-authors, we also evaluated the relevance of the hypotheses with Amazon Mechanical Turks, `GPT-3.5-turbo`, `Claude-v1.3`, and `GPT-4`. We report the results in Table 8 and found that the conclusion still robustly holds.

| Relevance Rater | w goal | w /o goal | $p$-value | spearmanr |
|---|---|---|---|---|
| Authors | 1.68 | 1.20 | $1 \times 10^{-10}$ | 1.00 |
| Turkers | 1.56 | 1.44 | $4 \times 10^{-2}$ | 0.10 |
| gpt-3.5-turbo | 1.05 | 0.94 | $5 \times 10^{-2}$ | 0.19 |
| claude-v1.3 | 1.18 | 0.92 | $2 \times 10^{-3}$ | 0.30 |
| gpt-4 | 1.49 | 1.12 | $1 \times 10^{-6}$ | 0.45 |

Table 8: We rated the relevance in the same way as Table 5. However, in this table we obtained the ratings not from the authors, but from four different evaluator types: Turkers, `gpt-3.5-turbo`, `claude-v1.3` and `gpt-4`. For each evaluator type, we calculate (1) the average rating of the candidate discovery when goal is (not) present in the proposers' prompt, (2) the $p$-value that the average rating when goal is present is higher under a t-test, and (3) the spearman rank correlation between its rating and the authors' rating. We find that the $p$-value is smaller than 0.05 in all cases, indicating that our conclusion is robust; additionally, more capable models has a higher correlation with the authors.

# 14   Meaningfulness: Relevance, Novelty, and Significance

Not every valid discovery is meaningful. For example, if the goal is to understand the topical differences between news from 2008 (Corpus A) and news from 2007 (Corpus B), the discovery that Corpus A "*contains news from 2008*" is completely valid by definition but meaningless, since it provides only trivial information and is irrelevant to the goal of understanding topical differences.

McGarry (2005) surveyed a list of desirable properties for discovery, and we condensed them into three submetrics to rate how meaningful a discovery is based on the exploration goal: 1) relevance, 2) novelty, and 3) significance. We evaluate these independently of validity and assume that the discovery is already valid. For example, the discovery that "something can travel faster than light" is meaningful if true, even though it is highly implausible.

We rate each submetric with ⓪, ①, or ②, where higher is better. We show the evaluation instructions below and present our rating on `text-davinci-003` proposed hypotheses.

## 14.1   Evaluation Instructions

**Relevance.** How relevant the discovery is to the goal. For example, suppose we were a student comparing essays rated as convincing vs. not convincing to figure out what writing style is convincing. Then:

- The discovery "*write in first person*" is directly related to the writing style, so we rate it ②.
- The discovery "*use the word "I"*", is not exactly a writing style, but can still inform the relevant underlying principle of "*write in first person*", so we rate it ①.
- The discovery "*argue for abortion*" does not tell us about the underlying writing style, so we rate it ⓪.

**Novelty.** The difficulty of generating the discovery, e.g. can we think of the discovery in 5 minutes with the goal but without looking at the corpora? For example, suppose we were an airline manager

trying to find improvements to the flight experience, and we were comparing negative reviews vs. positive reviews. Then:

- The discovery "*contain more negative language*" is almost certain for negative reviews, so we rate it ⓪.
- The discovery "*complain about the crew members*" is not entirely novel, but is not tautologically true and hence requires confirmation, so we rate it ①.
- The discovery "*mention a language barrier with the crew members*" is specific and hard to think of without looking at the data, so we rate it ②.

Note that our evaluation is "blinded to the samples": we still consider a discovery novel as long as it is hard to think of before looking at the corpora, even if it might be easy to think of after looking at the corpora. For example, the physical law that $F = ma$ is easy to observe if we have collected and plotted the data on acceleration, mass, and force; however, it might be difficult to think of before we see any such data, so we consider it novel.

**Significance.** Given the exploration goal, how beneficial is it to learn the discovery for the first time? For example, suppose we were an Amazon retailer trying to figure out what customers like and dislike about my product based on negative reviews and positive reviews. Then:

- The discovery "*accuses the team pushing out a bad product*" is not significant since it cannot direct the retailer to improve the product, so we rate it ⓪.
- The discovery "*asks for a more durable product*" gives some hints about how to improve the product, but isn't sufficiently helpful on its own, so we rate it ①.
- The discovery "*says the wrench is missing*" can lead to concrete actions for improvement, so we rate it ②.

## 14.2 Goal Leads to More Meaningful Hypotheses

|              | with-goal | no-goal | kappa | spearmanr | $p$ of avg           | worst $p$ of ind    |
|--------------|-----------|---------|-------|-----------|----------------------|---------------------|
| Relevance    | 1.68      | 1.20    | 0.56  | 0.71      | $1 \times 10^{-10}$  | $1 \times 10^{-8}$  |
| Novelty      | 1.24      | 0.97    | 0.37  | 0.50      | $5 \times 10^{-6}$   | $4 \times 10^{-2}$  |
| Significance | 1.56      | 1.05    | 0.46  | 0.64      | $2 \times 10^{-10}$  | $2 \times 10^{-7}$  |

Table 9: **Left.** For each metric, we report the average rating on hypotheses generated with or without using the exploration goal, and find that the former performs better. **Middle.** The inter-annotator agreement rate averaged across pairs of author evaluators, measured by Kappa and Spearman rank coefficient; we find substantial correlations between evaluators across all these subjective metrics, with relevance > significance > novelty. **Right.** We compute the $p$-values for the null hypothesis that "with-goal and no-goal result in the same performance". The $p$ of avg column reports the $p$-values after we average the ratings from all evaluators, while the "worst $p$ of ind" column takes the max of all $p$-values based on ratings of individual evaluators. Overall, the conclusions are statistically significant and they can be robustly reproduced across individual evaluators.

Compared to Zhong et al. (2022), we added the exploration goal to our prompt when generating hypotheses. Does this improve the quality of the proposed hypotheses? To investigate this, we sampled 100 problems from OPEND5 with distinct exploration goals and randomly sampled 2 hypotheses from GPT-3 with and without using exploration goal (see Figure 3), resulting in 400 hypotheses to evaluate. Three authors then rated their meaningfulness based on the three metrics defined in Section 3, while being blinded about which hypotheses were generated with the exploration goal.

The results are shown in Table 9. We found that, when prompted with the exploration goal, GPT-3 on average proposes more relevant, novel, and significant hypotheses; additionally, it proposes hypotheses with ratings higher than ⓪ 31%/21%/28% more often in terms of relevance/novelty/significance. Since this is a subjective evaluation, the Kappa inter-annotator agreement is only moderate, ranging from 0.37 to 0.56. However, we can still robustly conclude that the model can propose more meaningful hypotheses when conditioned on the goal: we calculate the $p$-values for the null hypothesis that with-goal and no-goal have equal performance, and we find $p$-values to be highly significant and robust across evaluators, for all three submetrics.

## 15 Full Pipeline of the Proposer

We present the full details of how we generated the hypotheses with the language model. The process roughly contains four stages: 1) obtaining representative samples for each corpus, 2) sampling hypotheses from GPT-3, 3) rewriting hypotheses, and 4) optionally plugging in example hypotheses.

**Obtaining representative samples.** This step is the same as Zhong et al. (2022), and we borrow the related text from that paper for the reader's convenience. Since $\mathcal{D}_A^{\text{res}}$ and $\mathcal{D}_B^{\text{res}}$ might overlap significantly, random samples from $\mathcal{D}_A^{\text{res}}$ and $\mathcal{D}_B^{\text{res}}$ might not be representative and informative enough for GPT-3 to notice the differences between the two distributions. Therefore, we choose samples that are representative of their differences. To find those samples, we fine-tune RoBERTa-Large Liu et al. (2019) to predict whether each sample comes from Corpus A or Corpus B and keep the top-p percentile samples with the highest confidence. Next, we take samples from the top-$p$ percentile to prompt GPT-3.

**Selecting samples to prompt GPT-3.** We randomly select $S =25$ samples from the top-5 percentile from Corpus A and Corpus B to prompt GPT-3 to propose the hypotheses, using the template shown in Figure 3 left. We require the length of the prompt to be at most 3,200 GPT-3 tokens (the max window size for GPT-3 text-davinci-003 is 4096) and gradually decrease the number of samples $S$ in the prompt until the prompt length is less than 3,200; additionally, we truncate each text samples to at most 256 GPT-3 tokens. Finally, to prevent GPT-3 from proposing hypotheses that reflect simple lexical correlations that can be detected with unigram models, e.g., *"uses the word "hey" more often."*, we incrementally construct the subset of samples for Corpus A and Corpus B such that at any time of the construction, no single word can appear $0.25S$ times more often in one corpus than the other. We repeat the same process for the top-20 and top-100 percentile until we obtain 60 hypotheses.

**Rewriting hypotheses with GPT-3.** As mentioned in Section 6.2, the hypotheses generated by GPT-3 are frequently statements about the corpus, while the validator requires the hypothesis to be a predicate on individual text samples. For example, when comparing definitions that people like from `UrbanDictionary.com` to other definitions, the hypothesis that the former *"is more likely to include slang or colloquial terms."* is a statement about a collection of text samples, rather than a predicate on an individual sample. $T(h, x)$ is undefined in this case, since it does not make sense to check whether a single text sample is more likely to include slang. Ideally, we want to detect these comparison statements and automatically remove the comparatives, e.g., rewrite it to *"includes slang or colloquial terms."*.

To detect and remove the comparatives from the hypotheses, we tag the part of speech for each word in the hypotheses using the NLTK package (Bird et al., 2009) and check whether any tag is JJR or RBR. If a hypothesis indeed contain theses tags, we prompt GPT-3 to rewrite the hypothesis. We show an example prompt in Figure 4.

**Plugging in example hypotheses (optionally).** We can also add a few problem-specific example hypotheses to the prompt to elicit more relevant hypotheses, and we do so by adding them to the "formatting instruction" part in the prompt used to propose hypotheses Figure 3. In OPEND5, we provided example hypotheses for each problem to steer our system to generate more meaningful discoveries; we produced the example hypotheses by prompting GPT-3 to generate a few hypotheses and selecting the meaningful ones from them.

For the reported discoveries in Section 6.1, we confirmed that they are unambiguously different from our provided hypotheses; otherwise, the system might have produced the discoveries by copying the provided hypotheses. We did not use the example hypotheses in Section 5 to test GPT-3's zero-shot understanding of the goal.

## 16 Collecting Data to Fine-tune the Validator

Here we provide a high-level description of how the data was collected. For each problem in OPEND5, we used our proposer to produce a list of hypotheses. We automatically judged each hypothesis on a subset of samples from the research split using GPT-3 text-davinci-002 (Ouyang et al., 2022), Flan-T5 (Chung et al., 2022), and a model trained with RLHF from Bai et al. (2022a). We created the input

Remove the comparatives. Remove mention of Group A or B if they appear.

Input: contain longer sentences than those from group B
Output: contain long sentences

Input: supports DACA more strongly
Output: supports DACA strongly

Input: uses the hashtag #AllLivesMatter more often
Output: uses the hashtag #AllLivesMatter

Input: is more likely to contain grammatical errors
Output: contain grammatical errors

Input: sounds happier than those from Group B
Output: sounds happy

Input: is more likely to include slang or colloquial terms
Output: includes slang or colloquial terms

Figure 4: The prompt to remove comparatives from a hypotheses.

distribution for training by combining and equally weighting the following $3 \times 2 = 4$ distributions: the subset of $(h, x)$ pairs that GPT-3/Flan-T5/"RLHF" considers Yes or No to be the most likely answer. We then collected averaged turker ratings for in total 3138 $(h, x)$ pairs and used them to fine-tune Flan-T5 to create the validator (Chung et al., 2022).

To test cross problem generalization capability of our D5 system, whenever we applied our D5 system to a problem in OPEND5 in Section 6.1, we used a validator that is NOT fine-tuned on the $(h, x)$ pairs from this problem. We achieved this by keeping track of which problem each $(h, x)$ pair comes from and split all the $(h, x)$ pairs into three folds based on the problems; whenever we applied our D5 system to a problem, we used the validator trained on the two folds that do not contain this problem.

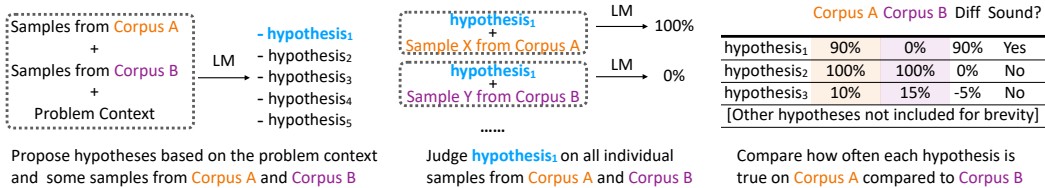

Figure 5: A sketch of the baseline method. The description can be seen in Section 4 and the actual prompts can be seen in Figure 3.

# 17 What Discoveries Did we Choose to Present

Our system in total produces 3296 discoveries on OPEND5. However, we do not have enough budget to validate every finding, since estimating $V$ is expensive (Section **??**). Therefore, from the remaining 3296 discoveries, we manually selected 21 discoveries that 1) the authors think are relevant enough, 2) are representative of potential use cases, 3) do not require expert knowledge for Turkers to judge, and 4) are likely to achieve a small $p$-value with fewer than 200 samples from $\mathcal{D}_A^{\text{val}}$ and $\mathcal{D}_B^{\text{val}}$. We then estimated their validity based on the procedure described in Section **??** by using fewer than 200 samples from the validation split and calculated the $p$-values.[3] Since we are testing multiple discoveries and each of them can be statistically significant merely due to chance, we keep 13 discoveries with $V$ that are significantly non-zero with $p$-value below 7%, a threshold determined by the Benjamini Hochberg's procedure with a false discovery rate of 10%. In other words, fewer than 10% of the discoveries presented are false discoveries in expectation.

---

[3]We determined the number of samples s.t. $V'$ can achieve a $p$-value of 0.005. Estimating $V$ for these discoveries costs $\sim$\$1500.

| discovery | V | p | V' | p' |
|---|---|---|---|---|
| argues for a path forward to promote the fair ... | 0.16 | 1.26e-04 | 0.35 | 2.01e-73 |
| refers to illegal immigrants as criminals | 0.09 | 6.17e-03 | 0.19 | 3.17e-38 |
| has an informal tone, such as slang or colloqu... | 0.08 | 2.35e-03 | 0.24 | 1.46e-35 |
| mentions lack of legroom | 0.16 | 1.15e-03 | 0.38 | 1.34e-45 |
| mentions children or family | 0.08 | 1.00e-05 | 0.11 | 8.05e-09 |
| Uses language that is positive or uplifting | 0.12 | 2.12e-03 | 0.24 | 4.18e-59 |
| references violence or aggression | 0.06 | 9.87e-03 | 0.17 | 4.25e-26 |
| involves physical activity, such as walking, p... | 0.13 | 4.92e-03 | 0.37 | 7.07e-101 |
| contains keywords related to business, finance... | 0.08 | 2.89e-02 | 0.35 | 1.45e-95 |
| mention disasters and crimes, such as plane ac... | 0.03 | 7.03e-02 | 0.09 | 4.61e-06 |
| discusses coronavirus-related topics | 0.21 | 1.01e-04 | 0.27 | 9.19e-78 |
| references pop culture, such as movies, books,... | 0.21 | 2.67e-04 | 0.58 | 2.09e-30 |
| uses vivid imagery and metaphors to convey a f... | 0.09 | 2.47e-02 | 0.45 | 5.04e-64 |

Table 10: The full table of discoveries, along with their $V$, $V'$, $p$, and $p'$ scores.

## 18 More Example Discoveries on OPEND5

**Analyzing errors in NLP systems.** We considered the task of perspectrum classification (Chen et al., 2019), which has the following instruction: "given a perspective and a claim, classify whether the given perspective supports or undermines the claim. If the perspective could possibly convince someone with different view, it is supporting, otherwise it is undermining." We considered two few-shot learning systems: GPT-3 Instruct Curie (Ouyang et al., 2022) and Tk-Instruct-11B (Wang et al., 2022). We focused on the perspectives where the ground truth label is undermining, and compare the following two corpora: Corpus A – the set of perspectives where Curie correctly classifies the input as undermining but Tk-11B is wrong, and Corpus B – the set where TK-11B is correct while Curie is wrong. We found that Corpus B more often "*Uses language that is positive or uplifting*" ($V \approx 0.12$, AUCROC $\approx 0.67$). One possible explanation is that Curie made many mistakes by misinterpreting undermining as a label for negative sentiment rather than a logical relation between the claim and the perspective.

**Comparing lyrics from different eras.** Compared to lyrics from the 70s, those from the 80s more often "*references violence or aggression*" ($V \approx 0.06$, AUCROC $\approx 0.58$).

**Describing distribution shift.** We compared the premises from the SNLI dataset and MNLI dataset, and the former "*involves physical activity, such as walking, playing, climbing, or biking*" ($V \approx 0.13$, AUC-ROC $\approx 0.64$). One possible explanation is that SNLI is based on image captions.

**Comparing discussion topics between bots and human users.** We compared the topical differences between tweets identified as written by bots vs. human users on Twitter, and our system finds that the bots more often "*contains keywords related to business, finance or trading*" ($V \approx 0.08$, AUC-ROC $\approx 0.61$). One possible explanation is that bots are frequently used to generate finance-related scams.

**Identifying temporal differences in news headlines.** We compared headlines published by ABC news across different years. Compared to 2014, headlines from 2010 "*mention disasters and crimes, such as plane accidents and assaults*" more often ($V \approx 0.03$, AUCROC $\approx 0.53$). Compared to year 2019, year 2020 more often "*discusses coronavirus-related topics*" ($V \approx 0.21$, AUCROC $\approx 0.65$).

**Describing text clusters.** We present two example descriptions for text clusters. One from Wikipedia: "*references pop culture, such as movies, books, and television shows*." ($V \approx 0.21$, AUC-ROC $\approx 0.73$); one from PoetryFoundation.com: "*uses vivid imagery and metaphors to convey a feeling*" ($V \approx 0.09$, AUC-ROC $\approx 0.65$).

## 19 Limitations and Future Work

We still face many challenges in building a broadly useful system. We describe technical challenges that machine learning researchers can tackle in Appendix 19.1 and organizational challenges that require domain experts in Appendix 19.2.

## 19.1 Engineering Challenges

**Hypotheses about the corpora might not be appropriate predicates on individual samples.** When comparing highly rated definitions from `UrbanDictionary.com` to others, our system generates the hypothesis that the former "*is more likely to include slang or colloquial terms.*" This is a statement about a collection of text samples, but the validator requires the hypothesis $h$ to be a predicate on individual text samples $x$. To address this, we used GPT-3 to automatically remove comparatives from the hypotheses, e.g. rewriting the hypothesis above to "*include slang or colloquial terms.*"

However, some versions of this problem were harder to remove. For example, when comparing reviews from American Airlines (AA) flights and Delta Airlines to understand which aspects of each airline are doing better/worse, the proposer generated the hypothesis "*mentions American Airlines' staff being unfriendly and unhelpful*". Interpreted literally, this hypothesis can only be true on the corpus of AA reviews, since it presupposes the review to be about AA. The correct predicate for use on individual samples should instead be "*mentions staff being unfriendly and unhelpful*" (without the words "*American Airlines*'"). Therefore, future systems should explicitly convert corpus-level statements to their corresponding correct predicates, and the metrics should evaluate whether the validity of the predicates implies the corpus-level statements.

**Beyond truth predicates.** Our work requires the discovery to be a truth predicate that maps a text sample to a truth value. However, scientific discoveries can be arbitrary natural language expressions; extending to more flexible expressions requires a significant redesign of our system and evaluation framework. Some more feasible near-term extensions include 1) allowing natural language expressions that map from text samples to real values, e.g., "how polite the sentence is compared to other samples from the corpora" or 2) using additional logical forms to combine individual truth predicates; e.g., learn a shallow and interpretable decision tree where each split point is a natural language predicate.

**Beyond corpus-level differences.** Our work focuses on describing corpus-level differences and validates a discovery by comparing how often it is true on each corpus. Future work can consider other ways to validate a discovery: for example, suppose each text sample is associated with a continuous target variable, we can validate whether a discovery is more likely true if the target variable is large.

**Investigating sensitivity towards prompt format.** In this paper we hand-crafted the prompt for the proposer and manually annotated the exploration goals on our own for OPEND5. However, due to budget limitation, we have not investigated how sensitive is our D5 system towards prompt formatting and paraphrasing, or whether the performance could have been improved with better prompts. Future works can investigate more in this research direction.

**Clarifying a discovery.** Some discoveries seem to have clear meanings on the surface, but they become ambiguous when we judge them on individual text samples. For example, judging whether a text sample $h$ = "*mentions people*" seems like an unambiguous task a priori; however, it is unclear whether it is true on the sample $x$ = "*I woke up this morning.*", since the "*people*" in $h$ is a plural form, while $x$ only mentions one person "*I*". Future work can use a language model to automatically clarify the meaning of a hypothesis and make it more specific, e.g., rewrite $h$ as "*mentions one or more humans.*"

**Correlation $\neq$ causation.** Like other tools that rely on correlations to analyze patterns in data (e.g., linear regression), our system cannot establish causal relations either. For example, when comparing self-reported happy moments from females and males, even if the former corpus has more samples that "*mention children and family*", it does not necessarily imply family plays a more important role in inter-personal relations for females; an alternative hypothesis is that females might mention any other people more often than males, hence leading to the observation that they mention family more often. Future work can use language models to propose what control hypothesis to test.

**Decreasing the cost of validation.** As alluded to in Section 3, estimating $V$ is extremely expensive as it requires a lot of human labor. Future work can consider an importance sampling procedure that uses $\hat{T}$ as a proposer to improve the sample efficiency of estimating $V$.

**Training a better proposer.** We developed a self-supervised learning algorithm to propose more valid hypotheses. However, it does not take into account the meaningfulness metric, and it is unclear how to manage its trade-offs with validity if they exist. We look forward to future works that can train a better proposer with as minimal supervision as possible.

**Combining Meaningfulness and Validity Metrics.** To simplify evaluation, we assumed meaningfulness to be independent of the magnitude validity $V$. Such an assumption allows us to directly evaluate hypotheses that are not necessarily valid but is also limiting for evaluating the final discoveries: for example, for that 2008 "*discuss economy*" more often than 2007, it would be way more significant if $V = 0.99$ compared to $V = 0.0000001$. Future works can propose better metrics that do not assume that validity and meaningfulness are independent.

**Extending to Non-English Language** OPEND5 is currently annotated with English goals and most of the corpora are in English. Future work can consider extending this to other languages.

### 19.2 Organizational Challenges

As discussed in Polanyi et al. (2000), it requires implicit community norms rather than explicit deductive logic to decide what counts as good research results; to guide our system to produce truly important discoveries, our system needs feedback from researchers who work in the domain of interest. However, except for machine learning, the authors do not have research expertise in most of the domains listed in Figure 2. We look forward to future contributions from other domains and list concrete directions below.

**What problems to solve?** We generated the problems in OPEND5 by reading relevant papers and guessing what domain experts might care about. However, our guesses can be inaccurate. Future works can directly gather problems from domain experts to reflect the actual usage of our system.

**How to interpret a discovery?** We asked for Turker's judgment to compute $T(h, x)$. However, many hypotheses require expert knowledge to interpret properly. For example, only law experts can reliably judge whether a contract $x$ satisfies the predicate $h$ "*contains a license grant that is irrevocable*." Domain experts are needed to evaluate the validity of a discovery and supervise the validator.

**What discoveries are meaningful?** Our work developed the evaluation instructions to approximately evaluate what hypotheses are meaningful. However, just as no one can become an outstanding peer reviewer simply by reading the review guideline, we do not consider it feasible to provide a gold evaluation simply by reading our instructions. Whether a discovery is meaningful depends heavily on implicit community norms, and we hope domain experts can provide better evaluation and training signals for our system.

## 20 Self-Supervised Learning with Open-Ended Problems: A Proof of Concept

Since the problems in OPEND5 are open-ended, our system could potentially produce discoveries with higher validity scores than our current system. Therefore, we design a self-supervised learning algorithm to improve an LM's ability to propose more valid hypotheses, using the principle that **it is easier to validate a discovery than to generate one**.

**Algorithm.** Suppose we are given a set of problems for training and an initial language model $m_{\text{init}}$. Our goal is to automatically generate a set of *prompt-completion* pairs to fine-tune $m_{\text{init}}$ so that it can propose hypotheses that are more valid. To generate a *prompt*, we randomly sample a problem and create a proposer prompt following the procedure in Section 4.1. To generate the desired *completion* given a prompt, we sample multiple hypotheses from $m_{\text{init}}$, approximate their $V'$ score on the samples in the proposer prompt with the same language model $m_{\text{init}}$ (Section 4.2), and select the highest scoring hypothesis. Finally, we use the prompt-completion pairs to fine-tune $m_{\text{init}}$.

However, since we cannot fine-tune instruction-tuned GPT-3, we can only experiment with Flan-T5 (Chung et al., 2022), an open-sourced instruction-tuned model that might only work well for easier "mini-problems". As a proof of concept, we tested our algorithms for describing groups of four samples, where each group comes from a text cluster. As an overly simplified example, we will give the LM the prompt "*Group A: 1. dog 2. cat 3. pig 4. cow. Group B: 1. phone 2. laptop 3. desk 4. cup*" as an input and the LM can output "*mentions an animal*" as a hypothesis.

**Data.** We created 33 corpora by merging all corpora in OPEND5 with the same domain, and automatically generated 4503 text clusters using RoBERTa embeddings (Aharoni & Goldberg, 2020). We focused on clustering because it can automatically generate a large amount of semantically coherent groups of samples. To create a pair of four samples, we randomly sampled a corpus, sampled two clusters within that corpus, and took four random samples from each cluster. To test

cross-corpus generalization, we reserved 28 of the 33 corpora to create mini-problems for evaluation, using the rest for training. We used Flan-T5 (Chung et al., 2022) as $m_{init}$ and sampled hypotheses with a temperature of 0.8. For training, we sampled 30,000 mini-problems and selected the best of eight hypotheses generated by $m_{init}$ as the target completion; for evaluation, we sampled 200 mini-problems to calculate $V$ with Turkers and 1500 mini-problems to calculate $V'$ automatically.

**Results.** We evaluated randomly sampled hypotheses from the language model before and after self-supervised training. The automated "self-evaluation" validity score $V'$ improves substantially from 0.22 to 0.37, and the "true" validity score $V$ according to Turker evaluation improves from 0.07 to 0.10, with a $p$-value of 0.02. This result provides preliminary evidence that our algorithm (or similar variants) could be applied to a large set of problems to improve the validity of the hypotheses; we expect future validators to simulate human judgments better, hence decreasing the approximated gap of improvement between $V$ and $V'$.

## 21   Comparing D5 to Naïve Bayes

We qualitatively compare the discovery generated by our D5 system to the top-5 unigram features extracted by Naive Bayes, a traditional exploratory analysis method. The Naive Bayes method is effective when the target difference can be saliently reflected by individual words. For example, "*yo*" implies a rap genre, "*die*" implies a language of Deutsch, and ["*rank*", "*higher*", "*univeristy*"] hints at the topic of "*college ranking changes*". Additionally, compared to black-box neural networks, such a method is fully interpretable.

In comparison, D5 can directly generate a semantically coherent description for the target difference, saving users' time to guess the underlying correlation by inspecting the top unigram features. Additionally, it can capture differences that are hard to detect at a word level; for example, "*the genre of biblical scripture*" is mainly reflected in its sentence structure rather than individual words. Finally, D5 only describes goal-related differences, while Naïve Bayes picks up on any discriminative feature; for example, when identifying the topical differences between a English and a Deutsch corpus, Naïve Bayes fails catastrophically and only picks up common determiners such as "*the*" or "*die*" instead of topic words, since they are the most useful feature at telling which sample comes from which corpus. Given the respective strength of D5 and traditional exploratory methods, we envision D5 to serve as a complementary method to traditional methods.

## 22   Annotation Interface to Collect Human-Generated Hypotheses

(This section describes an interesting research direction we did not have time to fully pursue.)

**Task.** To fine-tune the language model to propose better hypotheses and perform validation more accurately, we also designed an interface to collect human annotations earlier in the project. In this annotation task, the annotators see five text samples from each of the two corpora; they then write one or many natural language predicate(s) that describe how samples from the two groups are different and choose which text samples satisfy each predicate the annotator has written. Since it is challenging for humans to identify systematic differences between even groups of five sentences, we made the task easier for them by

- we chose the representative samples from each corpus to form the two groups of samples, similar to the process in Section 15, and

- we highlighted subspan of the text samples that are informative for how the two corpora differ. For example, if Corpus A is sports related while Corpus B is entertainment related, we hope to highlight sports-related words like "basketball". To automatically identify the text spans to highlight, we fine-tuned RoBERTa to classify whether a sample comes from Corpus A and Corpus B, used the SHAP library to calculate how much each text span influences the classifier's decision, and highlighted the text spans based on the influence.

A screenshot of the annotation interface can be seen in Figure 6.

**Preliminary Results** We performed initial experiments on text clusters formed on the wikitext-2 dataset (Merity et al., 2016). We asked the authors to write hypotheses for 30-50 samples and then compare the results with GPT-3 generated hypotheses. We found that human annotators were able to

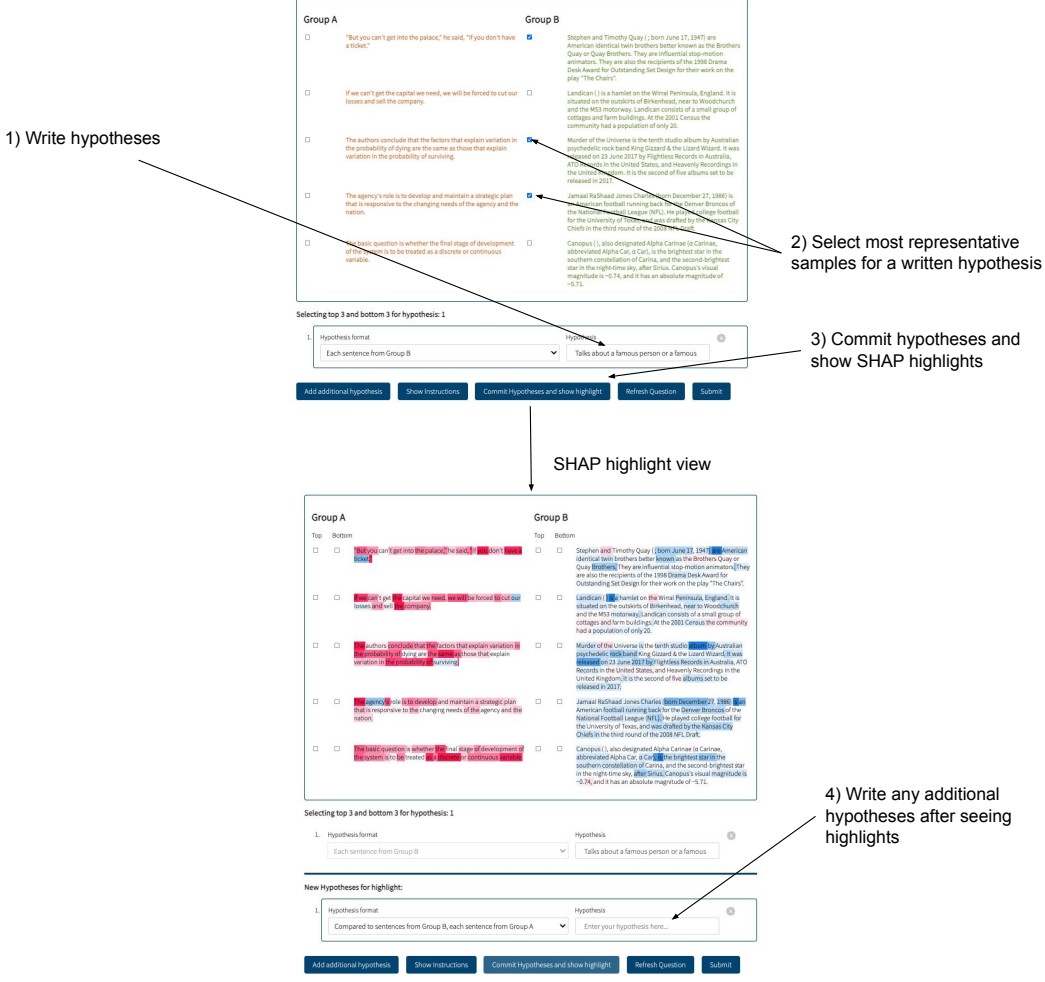

Figure 6: A detailed screenshot of our annotation interface.

write 2-4 valid hypotheses per pair of text groups, while GPT-3 text-davinci-003 was able to generate 4-6. Out of the valid generated hypotheses, approximately a third were variations on another valid hypothesis. The number of times humans were able to write a hypothesis that GPT-3 was unable to generate was around a third of the samples, while GPT-3 was able to generate a novel hypothesis humans have not thought about before in nearly every single text corpora. Given that GPT-3 is close to our author's ability to write hypotheses, we estimated that we would not be able to fine-tune T5 to propose better hypotheses with human annotations, and hence gave up on this research direction.

## 23 Datasets

Many of our datasets come from the following sources: the Computational Models of Social Meaning class from Columbia University`http://www1.cs.columbia.edu/~smara/teaching/S18/`, the ACL Anthology `https://aclanthology.org`, , and Kaggle datasets with an NLP tag. `https://www.kaggle.com`

**abc-headlines**. We collect headlines published by ABC news, an American news company from Kulkarni (2018). ABC headlines are directly downloaded from Harvard Dataverse. The year is extracted from the publication date field. Samples are constructed from the headline text. The data is downloadable from `https://doi.org/10.7910/DVN/SYBGZL` with license CC0 1.0.

**ad-transcripts**. We collect ad scripts from a variety of industries from Hartman (2019). Ad transcripts are directly downloaded from Kaggle. The top eight industries by frequency are selected. Newlines are replaced with spaces. The dataset is downloadable from `https://www.kaggle.com/datasets/kevinhartman0/advertisement-transcripts-from-various-industries` with license CC0 Public Domain.

**admin-statements**. We collect statements of administration policy from American presidents from Progress (2022). Administration statements are extracted from a collection hosted on GitHub. Extraneous symbols are removed and samples are split by paragraph. The dataset is downloadable from `https://github.com/unitedstates/statements-of-administration-policy#statements-of-administration-policy` and origin files have a Creative Commons Attribution 3.0 License.

**ai2-natural-instruction**. We collect a learning-from-instructions dataset released by the Allen Institute for AI from Mishra et al. (2022). Natural instruction tasks are directly downloaded without modification. The dataset is released under an Apache-2.0 license.

**airline-reviews**. We collect reviews of airlines collected from the review website Skytrax. Airline reviews for airlines, airports, and seats are downloaded from a public GitHub repository. Names of aircraft, airlines, countries, and traveler types are standardized. Ratings of 1, 4, or 5 on a scale of 5, and 1, 5, 8, or 10 on a scale of 10 are kept. This dataset can be downloaded via `https://github.com/quankiquanki/skytrax-reviews-dataset`.

**aita**. We collect posts on the "Am I The Asshole" Subreddit, an online forum people ask others whether they were in the wrong from O'Brien (2020). Posts from r/AmITheAsshole are downloaded from a praw scrape of Reddit. Topic areas are chosen based on common themes in posts and coarsely defined based on manual keywords. Each post can belong to multiple topic areas. The dataset can be downloaded at `https://doi.org/10.5281/zenodo.3677563`.

**all-the-news**. We collect news articles collected from various outlets between 2015 and 2017 from Thompson (2019). News articles are downloaded directly from the Components website. The titles are used as text samples.The dataset can be downloaded at `https://components.one/datasets/all-the-news-articles-dataset` .

**amazon-reviews**. We collect Amazon reviews collected from various product categories from Ni et al. (2019). Amazon reviews are downloaded from a 2018 crawl of the website. The first 100,000 review texts are treated as the text sample. The dataset can be downloaded at `https://nijianmo.github.io/amazon/index.html` .

**armenian-jobs**. We collect job postings in Armenia from Udacity (2017). The Armenian job postings dataset is downloaded from a snapshot on GitHub. Different IT jobs are manually coded and time intervals are defined in order to balance sample availability. The dataset can be downloaded at `https://www.kaggle.com/datasets/udacity/armenian-online-job-postings` .

**boolq**. We collect a reading comprehension dataset of yes/no questions from Clark et al. (2019). Boolean questions are downloaded directly as is. The dataset can be downloaded at `https://github.com/google-research-datasets/boolean-questions` with license CC-SA-3.0.

**clickbait-headlines**. We collect headlines across time from the Examiner, a clickbait news site from Kulkarni (2020a). The Examiner headlines are directly downloaded from Kaggle. The year is extracted from the publication date field. Samples are constructed from the headline text. The dataset can be downloaded at `https://www.kaggle.com/datasets/therohk/examine-the-examiner`, with license CC0: public domain.

**convincing-arguments**. We collect arguments on a variety of topics annotated for convincingness from Habernal & Gurevych (2016). Annotated arguments are downloaded from the GitHub repository. Arguments are sorted by rank. The bottom 400 are treated as "unconvincing", the top 200 are treated as "convincing", and the next 200 are treated as "somewhat convincing." The dataset can be downloaded at `https://github.com/UKPLab/acl2016-convincing-arguments`, with license CC-BY 4.0.

**craigslist-negotiations**. We collect dialogue from Craigslist negotiations, an online seller platform from He et al. (2018). Craigslist negotiations are downloaded from Huggingface. Sequences which contained a "quit" intention or "reject" intention are categorized as failures; those which contained an "accept" intention are categorized as successes. The mid-price is defined as the mean

price of the items sold. Within each category, the items are sorted by mid-price. The top half is treated as high-price and the bottom half is treated as low-price. This dataset can be downloaded at `https://huggingface.co/datasets/Hellisotherpeople/DebateSum` with MIT license.

**debate**. We collect evidence compiled for American competitive policy debate, published online by debate camps from Roush & Balaji (2020). The train split is downloaded from Huggingface. For each sample, we use the abstract as the text. Arguments are categorized by type, debate camp of origin, and topic/specific argument. For topics, we use domain knowledge to list relevant keywords for each topic and include any sample with a file name that includes any keyword. A single sample can belong to multiple topics. This dataset can be downloaded at `https://huggingface.co/datasets/Hellisotherpeople/DebateSum` with MIT license.

**dice-jobs**. We collect American technology job postings on dice.com from PromptCloud (2017). Job postings are downloaded from Kaggle. Posts from the six most popular companies are categorized by company. We remove miscellaneous characters and blank descriptions. We additionally apply our splitting procedure to reduce description length. This dataset can be downloaded at `https://www.kaggle.com/datasets/PromptCloudHQ/us-technology-jobs-on-dicecom` under CC BY-SA 4.0 .

**diplomacy-deception**. We collect dialogue from games of Diplomacy, which involves deception from Peskov et al. (2020). Diplomacy dialogues are downloaded from GitHub (all splits). The data are ASCII encoded and newlines are removed. Each message and label is treated as a sample. This dataset can be downloaded at `https://huggingface.co/datasets/diplomacy_detection` under unknown license.

**echr-decisions**. We collect facts of cases heard before the European Court of Human Rights from Chalkidis et al. (2019). Decisions are downloaded from a public archive. A random sample of 500 decisions is selected from the files. The samples with any violated articles are categorized as "violation," while the rest are categorized as "no violation." This dataset can be downloaded at `https://paperswithcode.com/dataset/echr` under unknown license.

**essay-scoring**. We collect essays from students from ess (2012). Essays are downloaded from a GitHub repository. Only essays from set 5 are considered. Essays with a score of at least 3 are categorized as good essays, while essays with a score less than 3 are bad essays. This dataset can be downloaded at `https://www.kaggle.com/c/asap-aes` under unknown license.

**fake-news**. We collect fake and legitimate news from Pérez-Rosas et al. (2017). Fake news articles are downloaded from the author's website. Full articles are treated as text snippets. This dataset can be downloaded at `http://web.eecs.umich.edu/~mihalcea/downloads.html#FakeNews` under CC-BY-4.0.

**fomc-speeches**. We collect Federal Open Market Committee (FOMC) speeches from 1996-2020, which describe Federal Reserve policy from Mish (2020). Fed speeches are downloaded from Kaggle. The macro indicator data are merged in on the year and month. Full speech text is split by paragraph and categorized by speaker, year, and macroeconomic indicator. This dataset can be downloaded at `https://www.kaggle.com/datasets/natanm/federal-reserve-governors-speeches-1996-2020` under unknown license.

**genius-lyrics**. We collect lyrics collected from Genius.com before 2020 from Lim & Benson (2021). Genius lyrics are downloaded from Google Drive. The lyrics are merged with song metadata and treated as samples. We categorize lyrics by hand-selecting popular artists, common genres, time periods, and view counts (over 1M views is high, 500k-1M is medium). This dataset can be downloaded at `https://www.cs.cornell.edu/~arb/data/genius-expertise/` under unknown license.

**happy-moments**. We collect self-reported happy moments and demographic characteristics from Asai et al. (2018). The HappyDB dataset is downloaded from the official GitHub repository. Demographic data is cleaned and merged into happy moments. Happy moment descriptions are treated as samples and are categorized by type of happy moment, country of origin, and other demographic features. This dataset can be downloaded at `https://github.com/megagonlabs/HappyDB` under unknown license.

**huff-post-headlines**. We collect headlines from the news outlet Huffington Post from Misra & Arora (2019) and Misra & Grover (2021). Huffington Post headlines are downloaded from Kaggle. The

short description of each article is treated as a sample and tokenized at the sentence level. This dataset can be downloaded at `https://rishabhmisra.github.io/publications/` under CC-BY-4.0.

**immigration-speeches**. We collect congressional and presidential speeches that mention immigration from 1880 to the present from Card et al. (2022). Immigration speeches are downloaded from the replication package. The speech text is preprocessed to remove extraneous spaces. We engineer features corresponding to time periods, well-known speakers, other significant time periods, the racial group under discussion, and the geographic area within the United States. This dataset can be downloaded at `https://github.com/dallascard/us-immigration-speeches/releases`.

**kickstarter**. We collect names of startups on kickstarter.com from Mouillé (2017). We download a 2018 crawl from Kickstarter from Kaggle. The project name is treated as the text sample. This dataset can be downloaded at `https://www.kaggle.com/datasets/kemical/kickstarter-projects?select=ks-projects-201612.csv` under CC BY-NC-SA 4.0.

**microedit-humor**. We collect funny sentences generated by making one-word edits to normal statements from Hossain et al. (2019). The Microedit dataset is downloaded from the author's website. We make the relevant edit to each text sample and treat the edited text sample as the data point. We bin the mean annotator grade into 4 and denote each as unfunny, neutral, funny, and very funny, respectively. This dataset can be downloaded at `https://paperswithcode.com/dataset/humicroedit`.

**mnli**. We collect a collection of sentence pairs annotated with textual entailment information from a range of genres from Williams et al. (2017). The MNLI corpus is downloaded from the official website. We treat the premise and hypothesis as text samples. This dataset can be downloaded from `https://cims.nyu.edu/~sbowman/multinli/`, most of which are under the OANC license.

**monster-jobs**. We collect American job postings on monster.com. Jobs on Monster.com are downloaded from Kaggle. Job descriptions are treated as samples and split at the paragraph and sentence level. We keep and categorize jobs from seventeen large cities. This dataset can be downloaded from `https://www.kaggle.com/datasets/PromptCloudHQ/us-jobs-on-monstercom` under CC BY-SA 4.0 .

**movie-tmdb**. We collect movie plot summaries from TMDB from Kaggle (2018). TMDB movie overviews are downloaded from Kaggle. We keep only English movies and bin popularity by deciles. The top decile is considered "hits," the 70-80th percentiles are considered "average," and the 30-40th percentiles are considered "bad." This dataset can be downloaded from `https://www.kaggle.com/datasets/tmdb/tmdb-movie-metadata21`.

**movie-wiki**. We collect movie plot summaries collected from Wikipedia from Robischon (2019). Wikipedia movie summaries are downloaded from Kaggle. This dataset can be downloaded from `https://www.kaggle.com/datasets/jrobischon/wikipedia-movie-plots` under CC BY-SA 4.0.

**news-popularity**. We collect news headlines posted on social media platforms from Moniz & Torgo (2018). Headlines are downloaded from a reproduction package. The headline and title text are cleaned, and the title is treated as the text sample. The 100 most positive and negative or popular and unpopular articles on each topic are used as distributions. This dataset can be downloaded from `https://archive.ics.uci.edu/ml/datasets/News+Popularity+in+Multiple+Social+Media+Platforms`.

**nli-benchmarks**. We collect training examples from various natural language inference (NLI) datasets from Liu et al. (2022). NLI benchmarks are downloaded from a public collection on Google Drive. We examine the premise and hypothesis separately as samples. This dataset can be downloaded from `https://github.com/alisawuffles/wanli`.

**npt-conferences**. We collect Non-Proliferation of Nuclear Weapons (NPT) conference transcripts from Barnum & Lo (2020). NPT conference notes are extracted from the accompanying replication package. Text is split by paragraph, and only paragraphs longer than 50 characters are preserved. Text is split into three time ranges: pre-2008, 2008-2012, and post-2012. This dataset can be downloaded from `https://journals.sagepub.com/doi/full/10.1177/0022343320960523`.

**open-deception**. We collect arbitrary lies and truths from any domain generated by crowdworkers from Pérez-Rosas & Mihalcea (2015). Open domain lies are downloaded from the public dataset

and lie texts are split into lies and truths. This dataset can be downloaded from `https://web.eecs.umich.edu/~mihalcea/downloads.html#OpenDeception`.

**open-review**. We collect submissions to ICLR, a machine learning conference from 2018 to 2021. Open review abstracts are accessed via the openreview API. We query for abstracts from the 2018-2021 ICLR blind submissions. Abstracts are classified based on rating: $>= 7$ ("great"), 5-6 ("good"), and $<= 4$ ("bad"). This dataset can be downloaded from `https://openreview.net/`.

**parenting-subreddits**. We collect posts from various parenting-related subreddits, which are text-based forums on the site Reddit from Gao et al. (2021). Posts from various subreddits are downloaded from the paper's GitHub repository. We clean the text and split the posts according to the topic(s) each post is tagged with. This dataset can be downloaded from `https://github.com/SALT-NLP/Parenting_OnlineUsage`.

**poetry**. We collect poems from PoetryFoundation.com from Bramhecha (2019). Poems are downloaded from a 2019 scrape of the PoetryFoundation website from Kaggle. The text is cleaned and split according to subject tags and authorship. This dataset can be downloaded from `https://www.kaggle.com/datasets/tgdivy/poetry-foundation-poems` under GNU Affero General Public License.

**political-ads**. We collect political ads observed by Facebook users from pol (2021). Ads are downloaded from the Ad Observer website, which maintains an aggregate of all collected ads. We extract targeting metadata from the targeting field and define splits according to age, gender, location, interests, time, and political lean. This dataset can be downloaded from `https://adobserver.org/ad-database/`.

**qqp**. We collect questions from Quora.com from Quora (2017).

**rate-my-prof**. We collect reviews of lecturers from RateMyProfessor.com from He (2020). We download a sample of RateMyProfessor.com reviews from an online repo. We clean the text and guess the gender of the reviewed lecturer from the first name using the gender-guesser package. Due to data availability, we consider only male and female names. To improve the quality of the classification, we remove any posts which use pronouns from the opposing sex (e.g. "him"). This dataset can be downloaded from `https://data.mendeley.com/datasets/fvtfjyvw7d/2` under CC BY 4.0 .

**radiology-diagnosis**. We collect impressions and medical histories of radiology patients from Pestian et al. (2007). Radiology diagnoses are downloaded from a GitHub copy of the original task dataset. We parse the metadata to retrieve the diagnostic code, decision type, impression, and patient history. Referencing the associated ICD codes, we convert codes to colloquial diagnoses (e.g. 786.2 denotes cough). We treat the histories and impressions as samples and split them according to diagnosis and level of consensus.

**reddit-humor**. We collect jokes posted on the Reddit forum r/Jokes, a message board for sharing jokes from Weller & Seppi (2020). Jokes are downloaded from the dev and test splits of the dataset. We clean the text and split the dataset according to whether they are labeled as funny. This dataset can be downloaded from `https://github.com/orionw/rJokesData` under Reddit License and Terms of Service, and users must follow the Reddit User Agreement and Privacy Policy, as well as remove any posts if asked to by the original user.

**reddit-stress**. We collect stress-related posts on Reddit from Turcan & McKeown (2019). We split the post text based on which subreddit they are posted on (related to PTSD, anxiety, or stress generally). Reddit posts are downloaded from `https://github.com/gillian850413/Insight_Stress_Analysis`, and we recommend following the Reddit User Agreement and Privacy Policy, as well as remove any posts if asked to by the original user.

**reuters-authorship**. We collect articles from various Reuters authors from Liu (2011). The articles are split according to the author. Reuters articles are downloaded from the UCI repository `https://archive.ics.uci.edu/ml/datasets/Reuter_50_50`.

**riddles**. We generated several riddles. The 3000 most common English words are manually copied from a website. Words with between 5 and 8 characters are kept. We create two popular riddles. First, we split words based on whether they have a duplicate character. We exclude any words with multiple "doubles" or more than 2 of any character. Second, we split words based on whether they have the letter T.

**scotus-cases**. We collect facts from cases heard by the Supreme Court of the United States (SCOTUS) from Alali et al. (2021). Supreme Court cases are downloaded from a GitHub repository. We identify state/federal parties by manually defining keywords. We split based on the winning party, the identity of each party, and the type of decision. We then define several time periods and relevant political eras and split decisions accordingly. Finally, we split according to the ruling's policy area and how it changes over time. The dataset can be downloaded from `https://paperswithcode.com/paper/justice-a-benchmark-dataset-for-supreme-court` under CC-BY-SA.

**short-answer-scoring**. We collect short answers from students from sho (2013). Short answers are downloaded from a GitHub mirror of the dataset. We consider only responses to essay set 1. The two scores are averaged and binned into good ($>= 2.5$), medium (1.5-2.5), and bad ($<1.5$). The dataset can be downloaded from `https://www.kaggle.com/c/asap-sas`.

**snli**. We collect a collection of sentence pairs annotated with textual entailment information from images from Bowman et al. (2015). The dataset can be downloaded from `https://nlp.stanford.edu/projects/snli/` under CC BY-SA 4.0.

**squad-v2**. We collect reading comprehension questions crowdsourced from Wikipedia articles from Rajpurkar et al. (2018). The dataset can be downloaded from `https://rajpurkar.github.io/SQuAD-explorer/` under CC BY-SA 4.0.

**stock-news**. We collect top news headlines on Reddit, an online message board from Sun (2017). Headlines are downloaded from a GitHub mirror. We clean the text and divide the samples based on whether the DOW rose or fell that day. The dataset can be downloaded from `https://github.com/ShravanChintha/Stock-Market-prediction-using-daily-news-headlines` under Reddit License and Terms of Service, and users must follow the Reddit User Agreement and Privacy Policy, as well as remove any posts if asked to by the original user.

**suicide-notes**. We collect posts from r/SuicideWatch and r/depression, two forums on Reddit fromHe (2021). The post title and body are combined to form the text samples. Samples are split based on whether they were posted in a suicide-related Subreddit. The dataset can be downloaded from a github: `https://github.com/hesamuel/goodbye_world`, under Reddit License and Terms of Service, and users must follow the Reddit User Agreement and Privacy Policy, as well as remove any posts if asked to by the original user.

**times-india-headlines**. We collect headlines from Times of India news from Kulkarni (2022). Headlines are downloaded from a Dataverse mirror. We use the first 1000 headlines in each year as samples. The dataset can be downloaded from `https://www.kaggle.com/datasets/therohk/india-headlines-news-dataset` under CC0 Public Domain.

**trial-deception**. We collect testimonies from witnesses in real trials from Pérez-Rosas et al. (2015). Trial testimonies are downloaded from the author's website. The testimonies are divided based on whether they are considered truthful. The dataset can be downloaded from `https://web.eecs.umich.edu/~mihalcea/downloads.html#RealLifeDeception`.

**un-debates**. We collect speeches from debates at the United Nations from Baturo et al. (2017). Debate transcripts are downloaded from the Dataverse reproduction package. Samples are divided based on the country and year of the snippet. First, we isolate samples from Russia, China, and the United States and specify 3 time periods of interest. Next, we divide all samples by the decade. Finally, we create distributions for 19 countries of interest. The dataset can be downloaded from `https://doi.org/10.7910/DVN/0TJX8Y` under CC0 1.0 .

**unhealthy-conversations**. We collect expert-annotated unhealthy conversations from Price et al. (2020). Conversation transcripts are downloaded from the official GitHub repository. For each annotated attribute, we split the dataset based on whether that form of unhealthy conversation is present in the sample. The dataset can be downloaded from `https://github.com/conversationai/unhealthy-conversations` under CC BY-NC-SA 4.0.

**urban-dictionary**. We collect definitions from UrbanDictionary.com, a crowdsourced English dictionary from Kulkarni (2020b). Urban Dictionary entries are downloaded from Kaggle. Definitions are split into groups representing the top 1, 5, and 10 percent of definitions ranked by both upvotes and downvotes; we sample 10,000 from each and create a control distribution by randomly sampling 10,000 definitions from all entries. The dataset can be downloaded from `https://www.kaggle.com/therohk/urban-dictionary-words-dataset` under CC0 Public Domain.

**wikitext**. We collect text snippets from Wikipedia from Merity et al. (2016). The Wikipedia snippets are loaded from HuggingFace. We remove any samples that are empty or start with '=' (which represent headings); samples are tokenized at the sentence level and used for clustering. The dataset can be downloaded from `https://huggingface.co/datasets/wikitext` under CC BY-SA 3.0.

**yc−startups**. We collect descriptions of companies that were part of the Y Combinator startup incubator from Bhalotia (2022). YCombinator company descriptions are downloaded from a 2022 scrape on GitHub. Only companies with long descriptions are preserved. Companies are split according to founder characteristics, year, "top company" designation, operating status, and location. The dataset can be downloaded from `https://www.kaggle.com/datasets/benhamner/y-combinator-companies`.

