# OpenReview forum: "Goal Driven Discovery of Distributional Differences via Language Descriptions"
_NeurIPS.cc/2023/Conference — NeurIPS 2023 poster_

### Official Review · Reviewer_vHRK · 2023-07-06

**Soundness:** 2 fair
**Presentation:** 2 fair
**Contribution:** 2 fair
**Rating:** 6
**Confidence:** 4

**Summary:**

This paper presents a new task called D5 (goal Driven Discovery of Differences between text Distributions via language Descriptions)
) where the objective is to identify and describe meaningful differences between two given text corpora. Authors created two datasets to evaluate their D5 task based: SYND5 and OPEND5, based on validity and relevance.

**Strengths:**

The creation of OPEND5 and SYND5 datasets for training and evaluation shows originality.

**Weaknesses:**

Task justification and applicability:
The paper introduces a new task D5  (goal Driven Discovery of Differences between text Distributions via language Descriptions). However, the need for such a task is not clearly articulated. The authors do not provide sufficient examples or scenarios where this task could be beneficial, making it difficult for the reader to fully appreciate the significance of this work. It would be beneficial if the authors could provide concrete real-world applications or problems where this task could contribute to a solution or add value. If the main usage is as the example shown in Figure 3, the users can use GPT to generate answers themselves nowadays.

Interpretation of discoveries:
The paper acknowledges that the interpretation of the discovered differences is challenging and often requires domain experts. However, the study doesn't suggest concrete ways to address this problem, which could limit the practical application of the D5 system.

Reliance on Turkers' judgments:
The paper relies heavily on the judgments of Mechanical Turkers to evaluate the validity of discoveries. This approach might introduce another layer of bias or inaccuracies due to the potential lack of expertise among Turkers as authors mentioned in the section 6.2.

Missing a conclusion section

Minor suggestion: the right edge of Figure 2 is cut off which need to adjustment.

**Questions:**

The above weakness are also my questions.

Evaluation Metrics:
The paper primarily reviews the task based on the metrics of validity and relevance, but also mentions evaluating subjective metrics like novelty and significance. Could the authors explain why validity and relevance were prioritized in the evaluation process? And can the authors provide a brief overview or key insights from the evaluations of novelty and significance, given that they are confined to the appendix due to space constraints?

Related work:
The author propose their contribution as the novel D5 task but also mentioned they mirrored Zhong et al. (2022)'s study. Could the author explain what is their novel contribution compared with Zhong et al's work?

**Limitations:**

The authors have indeed discussed several limitations of their work which they couldn't address at this stage.

---

> ### Author Rebuttal · Authors · 2023-08-09
>
> > "The authors do not provide sufficient examples or scenarios where this task could be beneficial, making it difficult for the reader to fully appreciate the significance of this work."
>
> The goal of the task is to explain how two corpora are different in a goal-driven way, and we provided examples of applications in the abstract, intro, Figure 1, and Section 6.1. Here is an example from the abstract: “The task input is a problem comprising a user-specified exploration goal (comparing the side effects of drug A and drug B”) and a corpus pair (collections of patients’ self-reported reactions after taking each drug). The output is a goal-relevant description (discovery) of how these corpora differ (patients taking drug A “mention feelings of paranoia” more often).” Such an automated system would be useful because exploring large corpora can generate useful discoveries but is time-consuming for humans.
>
> > “If the main usage is as the example shown in Figure 3 (naive prompting), the users can use GPT to generate answers themselves nowadays.”
>
> Naively prompting GPT with the goal alone would not work because the output needs to be grounded into the two given corpora, and using a validator is crucial to the performance. In Table 1 right gpt-4,w/g column, we find that prompting the language model (proposer) to generate an answer without validation has significantly lower performance (27% -> 8%). Intuitively, validator is crucial because 1) the language model proposer is not reliable to generate the correct answer within a single answer, and 2) while we cannot feed the entire corpus into the proposer due to context length limitations, we can feed individual samples from the entire corpus to the validator to filter out hypothesis.
>
> L298: Interpretation of D5’s output might require domain experts, thus limiting the impact of our application.
>
> We would like to clarify that our system is meant to be used in an assistive setting, where the users hold the ultimate responsibility to interpret the outputs of the system properly. Therefore, in practice we expect that domain expert users themselves to be able to interpret D5’s outputs in the domain of interest, or be calibrated on whether they can interpret the outputs.
>
> Why was novelty and significance evaluation presented in the Appendix?
>
> Indeed, we are constrained in space, and we prioritized presenting validity and relevance in the main paper because they are more objective in their definition. As you have recommended, we will include insights about evaluating novelty and significance in the main paper for the updated version.
>
> > "Could the author explain what is their novel contribution compared with Zhong et al's work?"
>
> Compared to Zhong et al.,
> - We collected a large dataset, OpenD5, so that we can study D5 in a data-driven approach. This allows us to identify more promising use cases and identify more limitations (Section 6).
> - We included an exploration goal in the task input, which can potentially lead to more useful output discoveries
>
> > “Missing a conclusion section”
>
> Conclusion section typically contextualizes results, discusses limitations, and looks forward. In our submission, sections 6 and 7 do this in detail. We consider it only a stylistic difference that we called our final sections “discussion and related work”.

---

> > ### Comment · Reviewer_vHRK · 2023-08-17
> >
> > Thank you for the response! After reading through the rebuttal, comments and other reviews, I have updated my score in the review.

---

### Official Review · Reviewer_sfxq · 2023-07-07

**Soundness:** 3 good
**Presentation:** 3 good
**Contribution:** 3 good
**Rating:** 6
**Confidence:** 3

**Summary:**

This paper proposes a goal-directed discovery and description of differences between text distributions (D5). Given two corpora and a research goal, the task is to generate a discovery in a natural language predicate. The paper creates datasets SynD5 and OpenD5, which can be used to evaluate  validity, relevance, novelty, and significance.


**Strengths:**

- The paper tackles an interesting task and formulates research discovery as an NLP task.
- The paper creates new SynD5 and OpenD5 datasets from a variety of problems.


**Weaknesses:**

- #1 The task is not well-formed or well-motivated.
- #2 The dataset quality is not ensured. As stated in LL119-123, OpenD5 does not ensure that there is ground-trugh and thus is not clear hwo reliable it is as a benchmark.
- #3 The paper is generally well-written but the key concept and main messages are hard to parse.

First of all, it is a great effort to compile hundreds of datasets into a single benchmark but I’m afraid that I’m not very clear what we can conclude after seeing method A perform better than method B on this dataset. After reading the paper, I wasn’t able to answer a question “What functionality assessment can this test bed be used for?” by myself.

Looking at examples in Figure 2, the OpenD5 data split looks synthetic and some of them rely on the ground-truth labels for the original task. (e.g., 10-star reviews and 0-star reviews.) In other words, how the original corpus was split into Corpus A and B is essential to answer the research question. Thus, if the split is given, the problem is partially solved already.

In my understanding, the task essentially asks if the model can figure out how those two corpora were created from the original corpus with additional context information as the research goal.

For Machine Learning problems, I believe analysis on each pair could be one research paper (e.g., the difference between SNLI and MNLI; LM’s output differences etc.) Then, again I’m not very sure what we can learn from evaluation results on the OpenD5 benchmark.

To me, the problem the paper is tackling is rather automating Exploratory Data Analysis using LLMs (like the paper below. To be clear, it’s considered concurrent work and the paper doesn’t have to compare with it.)

Pingchuan Ma, Rui Ding, Shuai Wang, Shi Han, Dongmei Zhang, Demonstration of InsightPilot: An LLM-Empowered Automated Data Exploration System, https://arxiv.org/abs/2304.00477


**Questions:**

- As raised in the Weaknesses above.
- Q. In 2.1 Task Format, is the goal description given for the task (and thus is included in the OpenD5 dataset)?


**Limitations:**

The paper discusses the limitations. For ethical concerns, I didn’t check the licenses of all the datasets (whether they can be redistributed after modification)

---

> ### Author Rebuttal · Authors · 2023-08-09
>
> Thanks for your review!
>
> > "The task is not well-formed or well-motivated …  the task essentially asks if the model can figure out how those two corpora were created from the original corpus with additional context information as the research goal."
>
> Just to clarify, this task is not about recovering how the two corpora were generated – quite the contrary, we actually tell the D5 system how the two corpora were generated (Figure 1 Samples from Corpus A include self-reported …). Instead, we want a D5 system to explain goal-related differences. For example, given Corpus A= 1-star reviews vs. Corpus B=10-star review and a goal of “understanding what makes the customer unhappy”, the output discovery “Corpus A is 1-star” is not relevant, and we hope to generate something like “Corpus A mentions delivery issues more often” (L141).
>
> > “What functionality assessment can this test bed be used for?”
>
> The benchmarks evaluate a D5 system’s ability to generate a goal-related natural language explanation for how two text corpora are different. This is different from Ma et. al, which operates on tabular data.
>
> > “Task Format, is the goal description given for the task (and thus is included in the OpenD5 dataset)?”
>
> The goal descriptions are also given in OpenD5.
>
> > "OpenD5 does not ensure that there is ground-truth and thus is not clear how reliable it is as a benchmark"
>
> Even though we do not have ground truth available, once a system produces a discovery, we can evaluate them with the metrics in Section 3. This is more reliable than assuming a fixed ground truth, as there could be multiple relevant discoveries and a single discovery could be phrased in multiple ways (similar to how in machine translation, assessing similarity to a “ground truth” translation is less reliable than assessing the quality of a translation directly).
>
> We also included a diagnostic evaluation, SynD5, for automatic evaluation that does have a single specified ground truth for each example.
>
> > License concerns for OpenD5
>
> The licenses for all 58 data sources are documented in Appendix 22. If there are any that violate a license, please let us know so that we can remove them.

---

> > ### Comment · Reviewer_sfxq · 2023-08-10
> >
> > Thank you for your response. I've read the response, and my concerns have been partially addressed. I want to keep my score as is.
> >
> > For the last comment about the license. At least, the following datasets are under unknown licenses, for which the author(s) should contact the owner/create for permission for modification and redistribution.
> >
> > - diplomacy-deception
> > - echr-decisions
> > - essay-scoring
> > - fomc-speeches
> > - genius-lyrics
> > - happy-moments
> > - immigration-speeches
> > - microedit-humor
> > - nli-benchmarks
> > - short-answer-scoring
> > - yc−startups
> >
> > For other datasets, for example
> >
> > > mnli
> >
> > Does it mean that OANC’s license should be included as part of D5?
> >
> > > open-deception
> >
> > The website says the software and data are distributed under GPL (the version not specified.) Does this mean that D5 has to be distributed under GPL? (poetry is also under GPL)
> >
> > > movie-tmdb
> > > https://www.kaggle.com/datasets/tmdb/tmdb-movie-metadata21 (LL1194-1195)
> >
> > The URL is incorrect in the first place.
> > https://www.kaggle.com/datasets/tmdb/tmdb-movie-metadata
> >
> > The website provides this description, which sounds problematic if it is redistributed.
> >
> > This dataset was generated from The Movie Database API. This product uses the TMDb API but is not endorsed or certified by TMDb.

---

> > > ### Author Response · Authors · 2023-08-10
> > > **Thanks for your response; we will provide downloading script to subsets of OpenD5**
> > >
> > > Thanks for your response and checking the license carefully.
> > >
> > > As you have recommended, we will reach out to the authors for datasets that have unspecified license; if they do not reply by the acceptance decision will be made, we will remove them from OpenD5 and **instead provide a fully automatic script to download the datasets**.
> > >
> > > Though not included in the supplementary material, we have already implemented the fully automated downloading script so that the license constraint would not hurt the accessibility of OpenD5.

---

### Official Review · Reviewer_EmG8 · 2023-07-07

**Soundness:** 2 fair
**Presentation:** 3 good
**Contribution:** 3 good
**Rating:** 3
**Confidence:** 3

**Summary:**

The paper introduces a new machine learning task - D5 - to model and discover differences between two large corpora by providing a task input as an exploration goal. The paper also introduces a synthetic dataset, SynD5,  for evaluation of D5, as well as OPEND5, a dataset containing 675 problems that D5 can be used to train on.

**Strengths:**

- The task tackled by the paper - generating discoveries based on corpora is impactful
- The paper provides in-depth description of their methodology and datasets, as well as how they were built
- The proposed framework is innovative

**Weaknesses:**

- OPEND5, the dataset that was built specifically for D5, has been collected manually by the authors, which could introduce biases.
- As the authors themselves mentioned, the extracted hypotheses can be merely correlations in the corpora and not necessary causations. However, D5 is built in such a way that it would treat them as 'discoveries'. Given the nuance around the types of things the model would extract, having clearer, more robust measures and datasets for evaluation of such a system should be in place before releasing it. Validity scores being quite low support this as well.
- The paper selects 21 out of 3296 examples to showcase as discoveries, potentially showing only cherry-picked examples.

**Questions:**

- I wasn't clear what was the purpose of the distractor (mentioned in 2.1)
- I’m not sure I understand the process described in 4.3 where a discovery’s validity is meant to improve by fine-tuning a LM on discoveries with validity scores obtained from an LM. If we can’t trust the validity of the LM to begin with, why would we trust a model fine tuned on it?

**Limitations:**

I've mentioned a couple limitations in the paragraphs above. I believe the authors have adequately managed to address the limitations in the paper.

---

> ### Author Rebuttal · Authors · 2023-08-09
>
> Thanks for your review!
>
> > “OPEND5 has been collected manually by the authors, which could introduce biases”
>
> We would like to note that for practical purposes, it is common practice in machine learning research for authors to manually aggregate datasets and annotate prompts themselves [2,3,4], especially for tasks that require significant background knowledge and do not have enough naturally occurring data online. We believe that our dataset meets the scientific standard as long as we are transparent and “present in-depth description of our datasets” (as your review has acknowledged). We take the stance of [2], that our performance might be improved with better prompts and hence represents a lower bound of what current systems can achieve. Future work can collect more authentic datasets to facilitate research in this area.
>
> > “Having clearer, more robust measures and datasets for evaluation of such a system should be in place before releasing it.”
>
> We would like to clarify that:
>
> - The main contribution of our paper is a new task, datasets, and a **baseline** system meant for **research** use, but NOT a robust system meant for broad adoption.
> - Our system is meant to be used in an assistive setting, where the users hold the ultimate responsibility to interpret the outputs of the system properly.
>
> Our submission only serves as a preliminary step towards a broadly applicable D5 system, and we agree that our system is limited in many ways. Therefore, we tried to document as many failure modes and limitations as possible and hope that future works can address them (Section 6.2 and Appendix 18). In the updated version, we will explicitly mention that our system is only meant for research use and should be used in an assistive setting.
>
> > D5 is limited because “correlation != causation”
>
> As mentioned above, our system is meant to be used in an assistive setting. Therefore, the users hold the responsibility to interpret D5 discoveries as correlations instead of causations, just as users of linear regression should be responsible for not interpreting the coefficients as causal.
>
> > "I wasn't clear what was the purpose of the distractor (mentioned in 2.1)"
>
> Corpora A and B will have two differences, one is the correct goal-related difference and the other is the distractor difference. If a system fails to use the goal, it might output the distractor difference. As a result, we can penalize systems that are not goal-driven. (L95)
>
> > “If we can’t trust the validity of the LM to begin with, why would we trust a model fine tuned on it?”
>
> The validator is trained to approximate human judgment (L187), and model-approximated human judgment is broadly used in the literature, e.g., RLHF[1]. Finally, we confirmed the success of our algorithm with human evaluators (L212), who are more trustworthy sources.
>
> > “The paper selects 21 out of 3296 examples to showcase as discoveries, potentially showing only cherry-picked examples.”
>
> We stated in Section 6.1 that this section is meant to be a qualitative evaluation that gives readers better insights about our current D5 system, and we agree that it can be potentially biased by the authors’ choice. However, our paper also includes a lot of quantitative evaluations, which are presented in Section 5.
>
> [1] Learning to summarize from human feedback
>
> [2] Language Models as Knowledge Bases?
>
> [3] Fine-tuned Language Models Are Zero-Shot Learners
>
> [4] Evaluating Large Language Models Trained on Code

---

> > ### Comment · Reviewer_EmG8 · 2023-08-21
> >
> > Thank you for the comprehensive responses - I appreciate the authors' efforts in addressing the potential limitations of the work, both in the paper and in the rebuttal. I still don't believe the task is well-formulated enough, and my main concern remains the potential of interpreting correlations as causations (or 'discoveries'), which can be easily misused. Because of this, I will be maintaining my score.

---

> > > ### Author Response · Authors · 2023-08-21
> > > **Thanks for your response!**
> > >
> > > Thanks for your response!
> > >
> > > We would like to restate that "Our system is meant to be assistive, where **the users hold the ultimate responsibility to interpret the discoveries as correlational rather than causal.**". As a result, we explicitly stated this limitation in the abstract and multiple other salient places in the paper (intro and discussion) so that most readers would notice it. Like any statistical modelling tools used in the machine learning literature (e.g., linear regression), insights learned by our system needs to be interpreted as correlations unless additional assumptions are made.

---

> ### Author Response · Authors · 2023-08-17
> **Any feedback on our author response?**
>
> Since the discussion period might end soon, it would be really helpful you can post any feedback for our author response so that we can know what confusions remain and how we can potentially improve our submission. Thanks a lot!

---

### Official Review · Reviewer_Eiwe · 2023-07-11

**Soundness:** 3 good
**Presentation:** 3 good
**Contribution:** 3 good
**Rating:** 7
**Confidence:** 2

**Summary:**

The authors formulate a new task, D5, that automatically discovers differences between two large corpora in a goal-driven way. The task input is a problem comprising a user-specified research goal (“comparing the side effects of drug A and drug”) and a corpus pair (two large collections of patients' self-reported reactions after taking each drug). The output is a goal-related description (discovery) of how these corpora differ (patients taking drug A “mention feelings of paranoia” more often).
In addition to the dataset the authors also build a D5 system, and to quantitatively evaluate its performance,
Finally, the authors discuss the limitations of the current D5 system, which discovers correlation rather than causation and has the potential to reinforce societal biases when misused.



**Strengths:**

1. The authors propose new tasks and datasets that are interesting and intriguing, which is considered a contribution to the community. The idea of generating discoveries with controllable goal is a valid and useful task.
2. Overall the task looks well-formulated, and the dataset is covering multiple data domains.
3. The writing and presentation is of high quality. The paper is generally easy to follow and understand.
4. The qualitative analysis part is very well done and complete.

**Weaknesses:**

1. More discussion could be done regarding the uneven distribution of data examples in different domains. For example only 10 from Health versus more than 200 in social sciences.
2. The authors designed a set of scores for evaluation metrics, and it would be better if there could be some sort of relevance study between the subjective scores with a set of human scorers.
3. Although using closed large language models are exciting because they generally provides better results, it would be nice to also add in any results with open-source models or smaller models, both for the sake of reproducibility (since OpenAI will deprecate the old APIs soon) and also to show the community where the current open models is at for your proposed tasks.

**Questions:**

Above.

**Limitations:**

Yes, the authors state limitations as well as possible biases.

---

> ### Author Rebuttal · Authors · 2023-08-09
>
> Thanks for appreciating our work!
>
> > (**New experiment**) “Would be nice to also add in any results with open-source models or smaller models”.
>
> Since some users might not have GPT-4 access and text-davinci-003 will deprecate soon, we followed your recommendations and ran a new experiment with gpt-3.5-turbo and flan-t5-xxl as the proposer on SynD5. Overall, 1) gpt-3.5-turbo delivers stronger performance than text-davinci-003, and 2) flan-t5-xxl is still far from the state of the art. These experiments require only 1 A100 GPU with 40 GB memory, or < $5 OpenAI API credit without any approval from OpenAI required. See Table 2 in the response pdf.
>
> We additionally trained a distilled 3B parameter validator that is similar in quality as flan-t5-xxl. We plan to release it so that people with a 24GB memory GPU can run our system.
>
> > "It would be better if there could be some sort of relevance study between the subjective scores with a set of human scorers"
>
> Here we present two types of correlations: 1) for the same metric, how well does the evaluator correlate with each other, and 2) how does different metric correlate with each other
>
> How well do evaluators correlate with each other: for each metric, we present the average correlation (Kappa or Spearman rank correlation) between evaluators in Table 6. We found that 1) evaluators’ judgements highly correlate with each other, 2) the correlation is the highest for the relevance metric, followed by significance and novelty.
>
> Correlation between different metrics: for each pair of authors, we compute the pairwise correlation between each metric, e.g., what’s the correlation between evaluator 2’s relevance rating and evaluator 1’s novelty rating. The metrics are substantially correlated, but still sufficiently different: for example, to predict the relevance rating of evaluator 1, it is more informative to use evaluator 1’s relevance rating rather than novelty rating.  See Table 4 in the response pdf for more details.
>
> > "Regarding the uneven distribution of data examples in different domains … for example only 10 from Health versus more than 200 in social sciences"
>
> For many health applications, the data is not publicly available due to privacy concerns. Meanwhile, many social science problems rely on online comments or public speeches, many of which are available online without re-distribution constraints.

---

> > ### Comment · Reviewer_Eiwe · 2023-08-12
> >
> > Thank you for your response. The new experiment is a good contribution added to the paper as well as a trained open source model. I keep my original score recommending accept of the paper.

---

### Official Review · Reviewer_cy6L · 2023-07-21

**Soundness:** 2 fair
**Presentation:** 3 good
**Contribution:** 2 fair
**Rating:** 6
**Confidence:** 4

**Summary:**

This work formulates a new task called D5, proposes a language model (LM) based approach to generate desired outputs in the context of this proposed task, and contributes a new dataset of open-ended problems to highlight how their LM-based approach can facilitate exploration of differences between two large English text corpora, with respect to a stated goal. The goals (for exploring differences) are stated in natural language, and the output 'discovery' of the differences is also generated in natural language.

The work shows how discoveries generated by their D5 system can be evaluated on both a synthetic diagnostic dataset (where ground truth differences are known and recovery of these differences can be directly tested), and on the open-ended OpenD5 dataset. The evaluation in the latter case depends on one somewhat objective criterion of validity, and then relevance to the user-stated discovery goal is evaluated manually using subjective human assessments, where humans are either the authors themselves or crowd workers (not domain experts). This work as a whole seeks to demonstrate the potential of LMs to facilitate human or subject matter exploration of hypotheses for differences in two corpora, and these discovered hypotheses are meaningful towards a user-stated goal.

**Strengths:**

The paper is well written and is clear in its definition of the proposed task as well as the objectives of the work. The goals carry real-world significance, as a valid and reliable method to enable exploration of goal-relevant differences between two corpora can help scientists and domain experts in generating useful hypotheses and potentially help generate insights that would have otherwise been too laborious or just not that obvious via a manual assessment (especially when the two corpora are big in size). The work takes a concrete step towards these goals of enabling human exploration and discovery. While exploratory differences between text corpora have been studied in NLP before, adding a goal, expressed in natural language, to this process of discovering differences and enabling systems to take this goal into account is a novel modification to this kind of exploration task. The OpenD5 dataset is a valuable data contribution if released and documented properly. Proposed evaluation criteria and robustness checks also serve as a key framework for designing similar tasks and making sense of the goals of the proposed system and task.

**Weaknesses:**

- While proposed criteria and robustness checks are valid and sensible, when the goal is facilitating human exploration and providing a system and framework to end-users, it is important that people other than the authors create the prompts or fill in the non-fixed portions of the prompt template because it is not clear if even slight differences in how the ‘open-ended’ parts of the prompt text (and its framing) could affect the generated hypotheses and discoveries. It is understandable that recruiting people to also generate prompts and assess discoveries for the same goal and input corpora but with prompts written by different people is difficult, especially if a well-powered human experiment is to be conducted. However, it is critical to study the effect of how the exploration goal is phrased — which includes paraphrases. While the validity and relevance to criteria are necessary and well-motivated eval criteria, arguably another really important criterion here (since the system is intended to help human exploration) is stability (which can map to intra-coder reliability): is system output robust to tiny, non-meaningful changes in the stated exploration goal?
In fact, this can be seen as an important component of system reliability – if we cannot expect consistently similar or the same discoveries for stated exploration goals or prompts that carry the same meaning and logically entail the same output, then the system is clearly not reliable for use by humans. Additionally, this robustness or stability is important to establish, given literature in NLP, for example, on adversarial robustness where minor perturbations in text (such as in text style which should be irrelevant to the task) could 'break' the NLP system, and more. See Qi et al (2021) for an example with text style transfer, Zhang et al (2020) for a relevant survey, and Wallace et al (2019) for some useful pointers on adversarial triggers in NLP.
In addition to this important robustness criterion, it is also generally important to study the effects and properties of the language of the exploration goal. The following questions can be used to create a more tested system of prompt design: What must be included? What are the constraints on phrasing? Does the order of specifying things matter? Is there a guide that should be followed when constructing the exploration goal for the most 'valid' and 'relevant' results? Do different phrasings of the same underlying exploration goal affect the validity or relevance of the model output discovery?
Even for the template portions of the prompt (which are not meant to be modified), it is not clear how this template was constructed and what happens when these templates are changed. Theoretically, could a different template achieve better results? Without such an analysis, both the usage of the system and the generalizability of the research work are negatively affected.

- One of the stated aims of this work and the proposed system is facilitating the discovery of unknowns or the generation of novel, previously unknown insights. However, including what has been presented in section 6.1, it is not clear at all if this has actually been established even using the OpenD5 dataset. With only the authors judging the discoveries and no subject matter experts involved, such a claim cannot be evaluated or it needs to be reframed and scoped out appropriately. Novel to the authors does not seem to represent a significant advance and does not seem to represent a step towards “discovering unknowns” stated in line 35. However, this is a critical potential contribution of this work and not taking concrete steps to establish this potential negatively affects the significance of the work. Also, in lines 284-285, the authors acknowledge that the signal about validity is unreliable, and since validity is critical when it comes to hypotheses generation and exploratory differences, it is unclear if the proposed framework is ready for uptake and if it could provide potential value to practitioners as things stand.

- The explicit acknowledgment and some discussion on some of the limitations are important and add to this work, however, the limitations that have been acknowledged are of great importance and represent harm. The exploratory analyses aspect and the aim of discovering the unknown make the harm of bias reinforcement even more critical, yet no path to mitigating these harms has been provided. The biases in authors’ subjective evaluations are also similarly really important here and it is not clear why at least other humans or recruited crowd workers cannot help in all the key human judgement-based evaluations, though the cost is certainly an understandable factor. Not incorporating domain experts and their knowledge in assessing discoveries is also a big limitation since the potential for this system to generate novel insights and find unknowns remains unclear. With the authors rightfully cautioning users about bias amplification and reinforcement towards the end of this paper, with no pathway to mitigate or understand when such cases arise, it is questionable whether the system should be used at all. There are also at least two other key limitations (discussed below) that also need to be addressed or at least comprehensively discussed in this work.

In summary, the key weaknesses of the work as it currently stands include a) system reliability for usage by end-users is not yet fully established, with more robustness checks on prompt construction and framing needed; b) while stated as a goal, the step taken in this work does not seem to demonstrate potential for generating novel insights or discovering unknowns, and evaluation signal for the validity of the discoveries that do get generated also seems to be unreliable; and c) limitations, both the ones explicitly acknowledged and the ones that are not, represent real-world harms, and the mitigation of these harms has not been tackled and as things stand, these harms seem to outweigh any benefits of the proposed system and framework.

References:
Fanchao Qi, Yangyi Chen, Xurui Zhang, Mukai Li, Zhiyuan Liu, and Maosong Sun. 2021. Mind the Style of Text! Adversarial and Backdoor Attacks Based on Text Style Transfer. In Proceedings of the 2021 Conference on Empirical Methods in Natural Language Processing, pages 4569–4580, Online and Punta Cana, Dominican Republic. Association for Computational Linguistics.

Zhang, W. E., Sheng, Q. Z., Alhazmi, A., & Li, C. (2020). Adversarial attacks on deep-learning models in natural language processing: A survey. ACM Transactions on Intelligent Systems and Technology (TIST), 11(3), 1-41.

Eric Wallace, Shi Feng, Nikhil Kandpal, Matt Gardner, and Sameer Singh. 2019. Universal Adversarial Triggers for Attacking and Analyzing NLP. In Proceedings of the 2019 Conference on Empirical Methods in Natural Language Processing and the 9th International Joint Conference on Natural Language Processing (EMNLP-IJCNLP), pages 2153–2162, Hong Kong, China. Association for Computational Linguistics.

**Questions:**

1. Is it safe to say that at least for the paper as it stands, “discovering unknowns” is not something that has been evaluated or judged? Is there a robust plan to enable this system to reliably help domain experts discover true unknowns?

2. What can we expect about the robustness of the system to variations in the phrasing and framing of the stated exploration goal by different non-author users (see the first point in the weaknesses section)? Also, from an end-user as well as a research perspective, are any experiments conducted to test what must be included in the exploration goal or other non-templatic parts of the prompt, what should be constraints on phrasing, if the order in which various things (about the corpus, goal, etc.) things matters? Can we expect that different phrasings of the same underlying exploration goal will affect the validity or relevance of the model output discovery?

3. How was the prompt template designed, and do we know if a different template could yield better results?

4. As noted above in the third point in the weaknesses section, even within the scope of identified limitations, it does not seem to be easily identifiable for an end user when correlation might get framed as causation or when bias reinforcement might happen, and therefore, can the current system be used with those potent limitations and harms as a possibility? In other words, given these known limitations and harms, can we currently handle them effectively, or if not, should D5 usage be held until those limitations are overcome or at least mitigated effectively?

5. Section 6.1 is an important assessment of OpenD5, however a lot of details do not seem to be provided which hampers the review but also the clarity a reader needs. The manual selection process of the 21 discoveries is not clear: how was the ‘representativeness’ of potential use cases judged? Also, how were criteria 3 and 4 actually assessed (lines 254-255)? Apart from this section, some other key details and justification behind certain decisions are not provided which are also important, these are noted below.


Suggestions (acknowledging possible space limitations for author response, questions above are separated from some questions below – addressing the below questions can improve the paper):

- With respect to lines 49-50, goal-relevance being higher is expected (and good that it does happen) since one model has goal-conditioning and the other was not designed to handle stated goals; but it is interesting that validity or correctness is higher too – analysis of what invalid differences are now not generated in the proposed model or what valid differences were missed in baseline model would be interesting (unless already done, in which case a clearer pointer to this analysis should be included).

- For the synthetic diagnostic test, it seems like the testing would benefit from either an additional or alternative approach to a synthetic test bed where the two corpora are not generated by language models (since systems being tested in this work are also language models, which calls into the question the strength of the proposed synthetic benchmark). Given that the properties for goal-relevant and distractor dimensions are around topic, genre, and language, a straightforward synthetic benchmark is using parallel corpora and then running actual topic models on them to generate and divide documents/texts by topic using the outputs of topic models: in this way, the corpora being compared remain real-world human-authored collections while the distraction vs goal-relevant test can still be conducted. OpenD5, while being a valuable real-world dataset contribution relevant to the task, does not test this distractor element present in SynD5. So something like the proposed alternative diagnostic with a) corpora authored by humans, and b) a known distractor element should provide a robust synthetic testbed arguably necessary to gain insight into the proposed framework for using LLMs.

- Line 98: cannot just stay Claude-v1.3 – what actually is this, exactly, needs to be specified.

- What courses on computational social science (line 110-111) were used? A list should ideally be provided in the appendix.

- Line 112: what is the inter-annotator agreement for the exploration goals?

- Line 179: why 60?

- For the hypothesis validator (lines 183-184), how good are language models in terms of simulating Turkers' validity judgement? Is this a valid proxy for this hypothesis validation step?

- Especially for synd5, it seems like there was an opportunity to test with a variety of open source open access models which is especially important since we should operate under the assumption that many users will not have accept to GPT-4.

- Line 238: It actually is important that non-authors rate relevance in order to get a more robust assessment of relevant improvement when using goals on OpenD5.

- Typo in line 294: the closing quotation marker should not be the beginning of a new line.

- Line 329 in related work section: topic models deserve a mention since they are used for the discovery of human interpretable latent concepts, and in fact relevant to one of the experiments conducted in this very work where authors use the topic as one of the dimensions varying between two input corpora samples (see the suggestion above around a more robust synthetic test).


**Limitations:**

While some key limitations are already acknowledged in this work, there does not seem to be a robust discussion on managing and mitigating these in real-world usage of the system. With things like correlation being potentially framed or interpreted as causality, for example, when can users expect such things to happen – without any markers or indications for a system output, it is hard not to be skeptical of any discovery generated by the proposed system. In other words, the key question that seems to be underexplored here: if it is not easily identifiable for an end user when correlation might get framed as causation or bias reinforcement is happening, should the system be used with those potent limitations and harms as a possibility, and given these known limitations and harms, if we do not know if they can be handled effectively or not, should D5 usage be held until those limitations are overcome or at least mitigated effectively?

Acknowledgment is an important system, but adequately addressing limitations needs more work, and there are also at least two more really important broader considerations not discussed in this paper that need to be addressed:

1. The work seems to operate in the English language alone, at least in terms of substantive testing and application in the paper itself. There is no discussion about theoretical generalizability to other languages and expectations around that, and at least there should be an acknowledgment of the lack of empirical generalizability of findings about the system to other languages. The languages in which the synthetic and contributed real-world dataset operates should be made clear by the end of the introduction itself.

2. Some important broader impacts/limitations are not being discussed and there should be more discussion in the paper about who gets to use the proposed system and potential double-use scenarios. For example, given that GPT-4 leads to better performance than the open-source alternative, if using this system for effective discovery relies on OpenAI API access and compute resources discussed in Section 8, many scientists and other end-users will probably not be able to use this method of exploration effectively, advantaging some communities over others — leading to many ramifications such as expanding existing inequities in access and resources. How can open-sourced tools be leveraged here, and can smaller models requiring fewer resources be incorporated effectively? This unequal opportunity question also ties to the previous point about language -- doctors, for example, can be operating in different cultural contexts using different languages to make their notes and we do not know if their experiences be used effectively for hypothesis generation same as those operating with English. Low-resource languages are also a potential limitation and concern that warrants discussion.

---

> ### Author Rebuttal · Authors · 2023-08-09
>
>
> Thanks for your thorough review!
>
> The weaknesses raised centered around (a) the societal impact of D5, (b) the robustness and reproducibility of our conclusions, and (c) our evaluation on novelty. We discuss (a) by clarifying the main goal of this submission, and discuss (c) by providing additional discussion
>
> We addressed (b) by adding three experiments following your recommendations:
> - We rated relevance with external non-authors (e.g crowdworkers)
> - To address concerns that some experiments depend on inaccessible models, we evaluated more accessible models (gpt-3.5 and flan-t5).
> - We constructed an extension of SynD5 using human-written texts.
>
> > Re: “Should D5 usage be held until limitations are mitigated?”
>
> We agree with this recommendation. Our main contribution is a new task, datasets, and a **baseline system** meant for research use, but NOT a system meant for productions.
>
> > Re: “Rating relevance with non-authors to obtain robust assessment” (new experiment)
>
> We chose to evaluate relevance ourselves, as crowdworkers might be untrustworthy [5,6]. However, we agree that external ratings would make our conclusion more convincing. Therefore, we used Amazon MTurks to test the conclusion that “prompting with goals leads to more relevant hypotheses'' (Table 2). Our conclusion is still significant, with a p-value of 0.04 calculated in the same way as in L243. Our conclusion also robustly holds when we rated relevance with three LM APIs: GPT-3.5-turbo, Claude-v1.3, and GPT-4. See Table 1 in the response pdf.
>
> > Re: “Test with open source open access models.” (new experiment)
>
> Since some users might not have GPT-4 access, we followed your recommendations and ran a new experiment with gpt-3.5-turbo and flan-t5-xxl as the proposer on SynD5. Overall, 1) gpt-3.5-turbo delivers stronger performance than text-davinci-003, and 2) flan-t5-xxl is still far from the state of the art. These experiments require only 1 A100 GPU with 40 GB memory, or < $5 OpenAI API credit without any approval from OpenAI required. See Table 2 in the response pdf.
>
> We additionally trained a distilled 3B parameter validator that is similar in quality as flan-t5-xxl. We plan to release it so that people with a 24GB memory GPU can run our system.
>
> > Re: “A synthetic test bed where the two corpora are not generated by language models.” (new experiment)
>
> We constructed an extension of SynD5 containing human-written texts by adapting the NYT dataset from [9], where each text sample is a New York Times article with a topic and a location label. The topic dimension has 9 different values (e.g., politics, arts) and the location dimension has 10 different values (French, Italy). We then constructed the SynD5 extension by following the same procedure in Section 2.2.
>
> We then evaluated several D5 systems on it. Similar to Table 2, we found that 1) LMs can leverage the goals effectively, 2) using a validator improves the performance. See Table 3 in the response pdf for more details.
>
> > Re: “Discovering unknowns” is not something that has been evaluated or judged?”
>
> We evaluated novelty in the following ways:
> - Proxy evaluation: we evaluated the novelty of the candidate discoveries to our best knowledge about the domain experts (results in Appendix 13.2). While we are not real users, the authors collectively have degrees in Econ, public policy, Linguistics, and Computer Science, which covers most of the scope of OpenD5 problems. Proxy evaluations are common practices in machine learning research, where the authors and the crowdworkers, rather than real users, evaluate the output of a machine learning system (e.g., asking crowdworkers rather than professional writers to evaluate creative story generation)[1].
> - Our judgment on novelty as real users with NLP expertise: We are experts in NLP research and are qualified to make novelty judgments for NLP-related OpenD5 problems. We and our colleagues (non-authors) consider the discoveries for “Analyzing errors in NLP systems” and “Describing distribution shifts” in (Section 6.1) and Appendix 17 to be novel.
>
> Indeed, the ideal evaluation is to 1) gather domain experts and 2) ask for their novelty rating. However, this is organizationally challenging (Appendix 18.2), and we do not consider this ideal evaluation feasible within the first paper written on this task.
>
> In the future, we hope this paper can attract enough attention from the community of interest and we are actively reaching out to potential real users. So far we have contacted the authors of [2, 3, 4]. All of them are surprised by the discoveries generated by our system; some of them found our D5-generated discoveries novel and are continuing working with us to explore how to use the D5 system properly.
>
> > Re: “It is critical to study the effect of how the exploration goal is phrased” and whether our system is robust to different templates
>
> We agree that our system might be sensitive to prompt templates; however, it is common for authors to manually write the prompts themselves [7,8]. As prompt robustness is an underlying issue for all NLP evaluations, evaluating and decreasing prompt sensitivity is generally considered its own research question, rather than intrinsic to each dataset.
>
> Again, thanks for your review, and we hope to hear back from you!
>
> [1] DOC: Improving Long Story Coherence With Detailed Outline Control
>
> [2] Accurate measures of vaccination and concerns of vaccine holdouts from web search logs
>
> [3] Human Heuristics for AI-Generated Language Are Flawed
>
> [4] Understanding the Usage of Online Media for Parenting from Infancy to Preschool At Scale
>
> [5] Artificial Artificial Artificial Intelligence: Crowd Workers Widely Use Large Language Models for Text Production Tasks
>
> [6] Crowdsourcing Beyond Annotation: Case Studies in Benchmark Data Collection
>
> [7] Language Models as Knowledge Bases?
>
> [8] Finetuned Language Models Are Zero-Shot Learners
>
> [9] Goal-Driven Explainable Clustering via Language Descriptions

---

> > ### Comment · Reviewer_cy6L · 2023-08-10
> >
> > I acknowledge I have read all of the authors' responses and rebuttals. The new experiments provide value, as does the new discussions and clarification on framing about the scope of this work (which should be incorporated in the paper). Most of my concerns and questions have been adequately addressed. The authors should discuss the new broader limitations pointed out in my review in the paper, or respond why it may not be the concern I think it is (English language only, closed-source models outperforming open-sourced ones) -- addressing them is out of the scope for this work and does not negatively impact the significance of this work, but a discussion could benefit readers and future researchers.
> >
> > One key concern that remains is the potential impact of framing the prompt. I agree that most works have the authors manually design the template themselves. That does not mean this is not a concern. Some discussion in appendix on the process of manually creating the template in this study could vastly benefit researchers, perhaps the different considerations themselves. It can be a very practical appendix section, either suggestions-based or a discussion of the experience when designing the template. But more importantly, other than the fixed template portions, what about the open-ended gaps that users of the systems have to fill in themselves? Even if robustness checks on the framing of those open parts cannot be conducted, suggestions or a guide to users on how to fill those gaps can help.
> >
> > Overall, the response, including new experimental effort, strengthens the work and I am increasing my score in my review.

---

> > > ### Comment · Reviewer_cy6L · 2023-08-10
> > >
> > > Please also note the minor suggestions in the original review that should ideally be addressed as well.

---

> > > > ### Author Response · Authors · 2023-08-10
> > > > **Thanks for your response**
> > > >
> > > > Thanks for your updated response. In the updated version of our paper, we will include
> > > >
> > > > - the process of designing the template and manually annotating the prompt, and how we reviewed internally to assure quality
> > > > - limitations that you have brought up, e.g., the dataset is mainly in English, the robustness to prompt template, etc
> > > > - other suggestions in the original reviews

---

### Author Rebuttal · Authors · 2023-08-09

Thanks to all the reviewers for their comments! We are glad that most reviewers agree that the task is well-motivated and potentially impactful, the paper is well-written, and OpenD5 is a valuable resource for the community.

Many of the weaknesses brought up by the reviewers were 1) limitations of the current system, and 2) whether the conclusion drawn from our paper is robust and reproducible by the broader community. We address 1) by clarifying the contribution of our paper and address 2) by **adding 3 experiments following the reviewers’ recommendations**.

> **Clarifying Our Contribution**

Many of the weaknesses brought up by the reviewers were in fact limitations that our submission has already discussed. We hypothesize that there might be a mismatch in expectation between us and the reviewers, so we would like to clarify that:

- The main contribution of our paper is a new task, datasets, and a **baseline** system meant for **research** use, but **NOT** a system meant for broad adoption.
- Our system is meant to be **assistive**, where the users hold the ultimate responsibility to use the outputs of the system properly.

Given that this is the first paper written on this task, we SHOULD expect the current system to have many limitations; in fact, one goal of collecting OpenD5 is to actively look for as many limitations as possible in a data-driven way (L59). Since these limitations might have never surfaced without our invested effort, **we consider our extended discussion about the limitations as our strength rather than weakness** (Section 6.2 and Appendix 18), and we hope that our discussion can open up future research that can address them.

> **New Experiments**

We followed the reviewers recommendations and ran three additional experiments (see our uploaded pdf for more detail):
- We rated relevance with external non-authors (e.g., human crowdworkers). Our conclusion about relevance robustly holds under 5 different types of evaluators.
- To address concerns that some of our experiments depend on models that are not broadly accessible, we evaluated more accessible models (gpt-3.5 and flan-t5) on SynD5. Now the experiments on SynD5 can be run with only a 24GB memory GPU and < $5 OpenAI API credit without any approval required.
- We constructed a version of SynD5 using human-written texts instead of model-written texts. We found that our conclusions in Table 1 unchanged.

> (New experiment 1) Rating relevance with non-authors

In our submission, we chose to evaluate relevance ourselves: even though it is an easy and common practice to recruit crowdworkers for evaluation, they might be noisy and untrustworthy [1,2]. To deliver the most reliable results, we chose to evaluate as many outputs as possible manually ourselves and ensured unbiasedness by being blind to how outputs were generated.

However, we agree that external relevance rating would make our conclusion more convincing. Therefore, we ran a new experiment and validated the conclusion that “prompting with goals leads to more relevant hypotheses'' (Table 2) by using Amazon Mechanical Turks. We still find our conclusion to be significant, with a p-value of 0.04 calculated in the same way described in L243. We ran additional robustness checks and our conclusion still holds by rating relevance with three language model APIs: GPT-3.5-turbo, Claude-v1.3, and GPT-4. Therefore, our conclusion on relevance robustly holds under 5 different types of evaluators, including expert authors, external crowdworkers, and language models from different companies with different levels of capabilities. See Table 1 in the response pdf for more details.

> (New experiment 2) Performance of Open Sourced Models on SynD5

To address concerns that some of our experiments depend on models that are not broadly accessible, we additionally evaluated gpt-3.5-turbo and flan-t5-xxl as the proposer on SynD5. Overall, we find that 1) gpt-3.5-turbo delivers stronger performance than text-davinci-003, and 2) flan-t5-xxl is still far from the state of the art. These experiments require only 1 A100 GPU with 40 GB memory, or < $5 OpenAI API credit.  See Table 2 in the response pdf for more details.

(We did not experiment with the LLaMa family since the instruction-tuned version is not yet available to us.)

> (New experiment 3) SynD5 with Human-Written Text

To address the concern that the text in SynD5 is model-generated, we constructed an extension of SynD5 with human-written texts by adapting the NYT dataset from [3], where each text sample is a New York Times article with a topic and a location label. The topic dimension has 9 different values (e.g., politics, arts) and the location dimension has 10 different values (French, Italy). We then followed the same procedure described in Section 2.2 to create this SynD5 extension.

We then evaluated several D5 systems on this dataset. Similar to the findings in Table 2, we found that 1) language models can leverage the goals effectively and 2) using a validator improves the performance. See Table 3 in the response pdf for more details.

[1] Artificial Artificial Artificial Intelligence: Crowd Workers Widely Use Large Language Models for Text Production Tasks

[2] Crowdsourcing Beyond Annotation: Case Studies in Benchmark Data Collection

[3] Goal-Driven Explainable Clustering via Language Descriptions

---

### Decision · Program_Chairs · 2023-09-21

**Decision:**

Accept (poster)

**Comment:**

This paper proposes an interesting new task, D5, and presents a dataset for the task. D5 is comparing two corpora and seeing how they differ for a specific goal. While some reviewers thought the motivation for this task was not clear, I believe the motivation is definitely there, and perhaps the authors can try to revise the paper to make that even clearer. Some other weaknesses were pointed out in the reviews, but they were mostly addressed well by the authors.

There was an ethics issue, mainly for how one would/could use the system, and whether bias and/or hallucination would present problems. The authors addressed the issue well, and I encourage the authors to revise the paper acordingly.

Overall, this is a good paper tackling an important and interesting problem.